# Molecular Screening Reveals De Novo Loss-of-Function *NR4A2* Variants in Saudi Children with Autism Spectrum Disorders: A Single-Center Study

**DOI:** 10.3390/ijms26125468

**Published:** 2025-06-07

**Authors:** Najwa M. Alharbi, Wejdan F. Baaboud, Heba Shawky, Aisha A. Alrofaidi, Reem M. Farsi, Khloud M. Algothmi, Shahira A. Hassoubah, Fatemah S. Basingab, Sheren A. Azhari, Mona G. Alharbi, Reham Yahya, Safiah Alhazmi

**Affiliations:** 1Faculty of Science, Department of Biological Sciences, King Abdul-Aziz University, Jeddah 21589, Saudi Arabia; w.baaboud@hotmail.com (W.F.B.); aalrofaidi@kau.edu.sa (A.A.A.); rfarsi@kau.edu.sa (R.M.F.); kalgothmi@kau.edu.sa (K.M.A.); shasouba@kau.edu.sa (S.A.H.); fbaseqab@kau.edu.sa (F.S.B.); azhari@kau.edu.sa (S.A.A.); mgalharbi@kau.edu.sa (M.G.A.); shalhazmi@kau.edu.sa (S.A.); 2Therapeutic Chemistry Department, Pharmaceutical Industries and Drug Research Institute, National Research Centre, Dokki, Cairo 12622, Egypt; 3Immunology Unit, King Fahad Medical Research Center, King Abdul-Aziz University, Jeddah 21589, Saudi Arabia; 4Department of Medical Microbiology, College of Science and Health Professions, King Saud Bin Abdul-Aziz University for Health Sciences, Riyadh 14611, Saudi Arabia; yahyar@ksau-hs.edu.sa; 5King Abdullah International Medical Research Center, Riyadh 11481, Saudi Arabia; 6Neuroscience and Geroscience Research Unit, King Fahd Medical Research Centre, King Abdul-Aziz University, Jeddah 80200, Saudi Arabia

**Keywords:** ASD, *NR4A2*, developmental disorders, speech impairment, recurrent variants

## Abstract

Dysregulated expression of nuclear receptor superfamily 4 group A member 2 (*NR4A2*) has recently been associated with autistic spectrum disorder (ASD), speech impairment, and neurodevelopmental delay (NDD); however, its precise role in the prevalence and etiopathogenesis of ASD has not been fully elucidated. Herein, we aimed to explore the role of *NR4A2* variants in the genetic underpinnings of ASD among Saudi children of different age ranges and phenotype severities. A total of 338 children with ASD from 315 unrelated families (293 simplex, 2 quads, and 1 quintet) were screened for *NR4A2* variants via exome sequencing (ES) of the genomic DNA extracted from peripheral blood mononuclear cells (PBMCs), after which the probands with identified *NR4A2* variants were further subjected to trio genetic analyses. ES analysis revealed 10 de novo *NR4A2* variants (5 indels/nonsense, 2 missense, and 3 variants affecting splicing) in 8 unrelated probands (2.37%) and 2 affected siblings from 8 unrelated families (6 simplex (2.04%) and 2 quads (8.7%)). Three *NR4A2* variants were notably recurrent among both affected and unaffected carriers. All identified indels and two splicing variants met the criteria for pathogenic/loss-of-function (LoF) variants according to the ACMG classification (PVS1), whereas the missense variants were classified as of uncertain significance (VUS). This study is among the first to identify such a high frequency of recurrent variants in an ASD cohort, suggesting their significant contribution to the etiopathogenesis of ASD within this population.

## 1. Introduction

Autism spectrum disorder (ASD) is a life-long developmental condition affecting 1% of children worldwide, as per the last report of the World Health Organization (WHO), and increasing global estimates highlight it as being one of the most urgent public health challenges [1]. The onset of ASD often occurs during early childhood and is characterized by the disproportionate development of social communication skills, as indicated by restricted interests and repetitive behaviors [2]. The core features of ASD are characterized by substantial heterogeneity, especially because they are often accompanied by various medical conditions that significantly impact the daily functioning of affected individuals, including intellectual disability (ID); language and/or motor impairment; gastrointestinal issues; obesity; sleeping problems; seizures; and psychobehavioral problems, including attention deficit and hyperactivity disorder (ADHD), social phobia, depression, and anxiety, among others [3].

While ASD diagnosis depends on clinical examination, many families with autistic children, particularly the firstborn, may encounter a “diagnostic odyssey” while seeking an explanation for their children’s condition [4]. Considering the high contribution of genetic factors to the etiopathogenesis of ASD, which are believed to account for ~ 30% of cases with ID and autism [5], and the estimates of high heritability reported in numerous family studies suggesting strong genetic contributions to ASD risk [6], patients are often subjected to first-tier clinical genetics testing for prognosis, recurrence risk assessment, and therapeutic intervention [7]. Owing to the emerging technologies of genetic analysis, including exome sequencing (ES), genome-wide microarrays, and whole-genome sequencing (WGS), the genetic architecture of ASD has become more elucidated, and hundreds of risk variants and genomic loci have been identified for both idiopathic and syndromic ASD during the last decade [8]. Among those, the nuclear receptor superfamily 4 group A member 2 (*NR4A2*) gene has recently been highlighted as a major candidate for causing ASD [9], whereas gene-disruptive mutations and chromosomal deletions that only overlap with the gene were implicated in a monogenic and consistent phenotype of neurodevelopmental disorders and language impairment, with or without seizures [10,11,12,13].

The *NR4A2* gene (nuclear receptor-related 1 (*Nurr1*)) is located on chromosome band 2q24.1 and encodes an orphan nuclear receptor that belongs to the nuclear steroid–thyroid hormone–retinoid receptor superfamily [14]. The encoded NR4A2 protein is widely expressed throughout different organs and peripheral blood; however, it is robustly expressed in the central nervous system (CNS), where it plays a central role in modulating the differentiation, maintenance, and survival of midbrain dopaminergic neurons [15]. The composition and physiological role of NR4A2 as a ligand-independent nuclear receptor provide insight into the heterogeneous phenotypes reported in ASD patients with *NR4A2* variants. The NR4A2 protein consists of two highly conserved functional domains: a DNA binding domain (DBD), which includes two C4-type zinc fingers, and a C-terminal ligand binding domain (LBD), through which NR4A2 binds to specific motifs in DNA hormone response elements to induce constitutive transcription [16]. Therefore, a mutated *NR4A2* gene may encode a misfolded protein or a dysfunctional DBD or LBD, which leads to impaired/loss of function [12]. Moreover, the tolerance landscape of *NR4A2* has shown that the zinc finger domain in the DBD is a particularly intolerant relative to other regions in the protein [13], which explains the deleterious effect predicted for mutations occurring in this region.

In recent years, there has been a noticeable increase in population-based studies that identify ASD susceptibility genes in several populations worldwide [17,18]; however, few epidemiological studies have scrutinized the “true” prevalence and genetic risk of ASD in Middle Eastern populations [19]. In a recent report by AlBatti et al. [20], the prevalence of ASD in one city in Saudi Arabia was estimated to be 2.51%. In light of this high prevalence and the existing links between NR4A2 and dopaminergic neurodevelopment, this study aimed to explore the role of *NR4A2* LoF variants in the genetic underpinnings of ASD among Saudi children of different age ranges and varying phenotypes.

## 2. Results

### 2.1. General Demographic and Clinical Phenotypes of the Study Cohort

As summarized in Table 1, the ages of the children with ASD ranged from 3 to 14 years, with a median age of 8 ± 2.684 years, and the distribution across males and females was non-significant. The intellectual profiling of the ASD cohort revealed that the majority (73.96%) had mild intellectual disability (ID), with FSIQ scores ranging between 55 and 126, whereas 21.3% and 4.74% of patients had moderate and severe ID, respectively, with FSIQ scores ranging between 19 and 54 and a median FSIQ score of 84 ± 24.55 for the total cohort. Concerning the IQ subscales, the children with ASD had significantly higher NVIQ scores than verbal (VIQ) scores (*p* = 0.0139). The same was observed when the Vineland scores of patients aged <4.5 years were compared with those of patients aged >4.5 years (*p* < 0.0001). The distributions of FSIQ, VIQ, and NVIQ scores were significantly different among different age ranges (*p* = 0.0216, 0.0098, and 0.0446, respectively), with the highest scores observed in younger patients, unlike the Vineland, PPVT, and SRS scores, which showed the same distribution pattern in favor of younger patients; however, no significant difference was observed (Appendix A). The VIQ and Vineland scores were significantly different between male and female patients (*p* = 0.0184 and 0.0242, respectively), with the latter showing higher scores on both subscales and a general pattern of higher FSIQ, NVIQ, PPVT, and SRS scores despite the absence of a significant difference. The children with ASD displayed varying degrees of global developmental delay (DD), with mostly mild-to-moderate intellectual disability (ID), which was concurrent with a wide spectrum of comorbidities. Gastrointestinal issues were the most common conditions observed among the affected children, followed by sleeping disorders, language/speech impairments of different degrees, and motor and feeding difficulties, which had convergent frequencies. Other clinical features observed included short stature and autoimmune disorders, whereas facial dimorphism was the least observed phenotype. Seizures of different types were also observed in 35 patients (10.35%), while the remaining patients were/became seizure-free after treatment with appropriate antiepileptics. Psychobehavioral problems, including hyperactivity/ADHD and anxiety, were the most frequently observed problems among the children with ASD, followed by sensory processing difficulties, which included hyper/hyposensitivity to sensory stimuli such as light, pain, and noise, and learning difficulties, which were observed in 115 patients (34.02%) in the ASD cohort. Attachment disorders and obsessive–compulsive traits were observed at lower frequencies across the cohort, whereas aggressive behavior was observed in only 15 patients (4.43%).

### 2.2. Genetic Analysis: Frequency of De Novo NR4A2 Variants Among Children with ASD

The overview of the study design and genetic analysis workflow for children with ASD is demonstrated in Figure 1. The exome sequence analysis identified 10 de novo cis-heterozygous *NR4A2* variants (5 indels/nonsense, 2 missense, and 3 at a splice region), which may be related to the clinical phenotypes observed in 8 unrelated probands (2.37%) and 2 affected siblings [n = 10, 8 males (80%; 3.28% of the total cohort), and 2 females (20%; 2.13% of the total cohort)] from 8 unrelated families (6 simplex (2.04%) and 2 quads (8.7%)). The median age ranges of the *NR4A2* probands and their affected siblings were 8 ± 2.644 and 7 ± 2.828 years, respectively, with no significant differences observed. All variants were validated by Sanger sequencing, which revealed the recurrence of two variants in two affected siblings of unrelated probands. No significant difference was observed in the variant distribution between male and female patients. As detailed in Table 2, each proband carried at least two *NR4A2* mutations. The first proband (A1) carried multiple mutations that included an indel (c.-2-2del) located in intron 2, and two nonsense variants (c.1del (p.M1*) and c.548del (p.P183*)) located in the regulatory N-terminus domain (NTD) in exon 3, both of which were expected to trigger premature termination (Appendix A). Another five probands (A2, A6, A11, A27, and A38) carried an indel (c.44_45insA, p.S16*) that introduced an unexpected stop codon after 41 amino acids in the NTD region (Appendix A), concurrent with a frameshift/splice-acceptor CNV (c.1159-81_1540+67del, p.F387*) generated by a chromosomal deletion of 919 bp from the ligand-binding domain (LBD), encompassing exons 6 and 7 (203 and 179 bp, respectively) and the flanking region from intron 5 (position 850 of 930) to intron 7 (position 67 of 149). Two of the five probands (A27 and A38) also carried a missense variant (c.537G>C, p.K179N) and a synonymous variant (c.39A>G, p.Q13, respectively). The c.44_45insA nonsense variant was similarly identified in another proband (A3), who also carried another indel (c.536del (p.K179*)) (Appendix A), whereas the CNV was identified in the 8th proband (A12), with a missense variant (c.5_6delCTinsTG, p.P2L) and a nonsense variant (c.534del, p.F178*) in the NTD region, both of which were expected to trigger premature termination (Appendix A).

### 2.3. Molecular Screening of Concordant and Discordant Siblings Reveals Deleterious NR4A2 Variants

The genetic analysis of affected siblings (Table 2) revealed that one concordant sibling [ID: AS_12] did not share any of the de novo NR4A2 variants identified in his older sister (proband: A12) but carried an intronic indel (c.-2-8del) located in intron 2 and the c.1del nonsense variant of the NTD (Figure 2A). The other concordant sibling [ID: AS_27] shared the nonsense variant c.44_45insA with his older brother (proband: A27), which was inherited from a healthy mother, in addition to the c.1del variant that occurred de novo in this sibling (Figure 2B).

All affected NR4A2 subjects (probands and siblings) tested normal for fragile X syndrome and common ASD/NDD-associated genetic variants. The discordant (unaffected) siblings were also examined to determine whether they share any putative risk alleles with their affected siblings. Among the 14 siblings tested, 5 (35.72%) carried different missense or recurrent/de novo nonsense NR4A2 variants located within the NTD region of the protein (Table 3). The discordant sibling of the first quad (B12) did not share any of the variants identified in his older siblings (A12 and AS_12) (Figure 2A) but carried a de novo missense variant (c.536_537delCTinsGC, p.K179S). The discordant sibling of the second quad (B27) shared the nonsense variant c.44_45insA with her older brothers (A27 and AS_27) and carried the missense variant (c.5_6delCTinsTG, p.P2L) that occurred de novo in this sibling (Figure 2B). The remaining discordant siblings from simplex families (B3_2 and B11_2) carried de novo insertion variants (c.30_31insG, p.S11* and c.14_15insT, p.Q5*, respectively) that were absent in their affected siblings (Figure 2C,D), whereas the 5th discordant sibling (B38) shared the synonymous variant (c.39A>G, p.Q13) with her proband (A38).

### 2.4. Functional Consequences of the Identified NR4A2 Variants

All identified NR4A2 variants were absent in the gnomAD and ExAC databases, and no clinical significance assessments were submitted for these variants in ClinVar. As shown in Table 2 and Table 3, all the indels identified in the probands and their concordant/discordant siblings met the criteria of pathogenic (loss-of-function (LoF)) variants according to the ACMG classification (PVS1), with 10 ACMG points (10P and 0B). The seven indels were located in a pathogenic variant-enriched NR4A2 region, in which 22 pathogenic variants were previously identified, but none were predicted to undergo nonsense-mediated mRNA decay (NMD) given their location <100 nucleotides from the start codon. The three missense variants observed in all NR4A2 subjects (both affected and unaffected) were classified as of uncertain significance (VUS) with 2/3 ACMG points (2/3P and 0B), whereas the synonymous c.39A>G was classified as likely benign. Despite the VUS classification of the c.5_6delCTinsTG missense variant, it was identified as “deleterious” and “probably damaging”, according to the SIFT and PolyPhen scores (0 and 0.929, respectively). Regarding the splice region variants, the SpliceAI Lookup tool predicted that the intronic c.-2-8del variant was located within the regulatory homopolyermic region (i.e., 8 bp upstream of the canonical splice site); therefore, it was predicted to have no significant impact on normal splicing. Meanwhile, both the c.-2-2del and c.1159-81_1540+67del variants are predicted to cause frameshift exon skipping, which prompts NMD-mediated loss of function. The intronic c.-2-2del indel displayed the highest predictive scores for a detrimental effect (12 ACMG points:12P and 0B), where a cryptic splice site was detected with a MaxEntScan score of 6.2 (offset of 17), causing splice acceptor loss 1 bp upstream of the variant at the pre-mRNA level (delta score of 0.9) and splice donor loss 18 bp upstream of the variant at the pre-mRNA level (delta score of 0.49) in the variant allele, which indicated impaired splicing of exon 3 in the probands carrying the variant. Similarly, the CNV (c.1159-81_1540+67del) was predicted as being a LoF variant with 10 ACMG points (10P and 0B) as it contains two breakpoints in the same loss-of-function-causing gene and a coding region of a loss-of-function variant that is ultimately expected to affect the haploinsufficient NR4A2 gene. No cryptic splicing site was detected in this variant, but the SpliceAI Lookup prediction revealed a splice acceptor/donor loss 1 bp upstream from the variant (delta scores of 0.9 and 0.92, respectively) and a splice acceptor/donor gain exactly at the deleted base and 2 bp upstream from the variant, respectively (delta scores of 0.84 and 0.3, respectively). These results were supported by RT–PCR followed by targeted PCR amplification of the region flanking the deleted exons, which revealed the absence of the expected wild-type transcript at the expected size (~540 bp) in amplicons obtained from all six CNV carriers compared with those obtained from concordant and discordant sibling controls or other probands who do not carry this variant (Figure 3), which indicates the potential role of this variant in disrupting normal splicing. Importantly, the absence of the canonical transcript in all CNV-positive samples suggests that the variant exerts a deleterious effect on splicing. The intellectual profiles of NR4A2 subjects are detailed in Appendix A.

### 2.5. Clinical Phenotypes and Intellectual Profiles of NR4A2 Subjects

The clinical phenotypes observed in the affected *NR4A2* subjects were prominently related to disproportionate language/speech and motor development, where all of the variant-carrying children (100%) had different degrees of language/speech impairment, ranging from delayed echolalia and speech apraxia/dyslexia to absent speech, concurrent with different types of movement disorders, including motor tics, dystonic posturing and/or choreoathetoid movements, and ataxic gaits (Table 2). The second most common phenotype observed among the affected *NR4A2* subjects included gastrointestinal problems (70%), followed by sleeping issues (60%), and hypotonia (50%). Seizures of varying degrees were observed in four patients (25%), and the least frequent phenotypes were facial dimorphism and Ehlers–Danlos syndrome (EDS), concomitant with cutaneous hyperextensibility and scoliosis, which were each observed in one patient (10%). The most common psychiatric issues observed in affected carriers included hyperactivity and anxiety (40% for each), followed by attachment disorders (30%), and sensory processing issues (20%).

Obsessive–compulsive disorder (OCD) and aggressive behavior were each observed in only one patient (10%). The intellectual profiling of affected carriers revealed a median FSIQ score of 63 ± 8.842 (range: 50–81), with two patients (20%) having borderline scores (>70), six patients (60%) having mild ID (IQ = 60–68), and two patients (20%) having moderate ID (IQ = 50–54). The mean FSIQ of affected carriers was significantly lower (*p* < 0.0001) than that of carrier and non-carrier discordant siblings (Figure 4A). The same was observed for the scores of the VIQ (Figure 4B), NVIQ (Figure 4C), Vineland (Figure 4D), PPVT (Figure 4E), and SRS (Figure 4F) subscales (*p* < 0.0001/for all), with no significant differences observed between the carrier and non-carrier discordant siblings. The limited number of *NR4A2* subjects did not allow an age-equivalent stratification for Vineland scores; however, the mean age for all discordant siblings, including both carriers and non-carriers, was 5.5 ± 2.345 (range: 3–10), which falls into the 4.5-years-old age equivalency of Vineland scores.

## 3. Discussion

Undoubtedly, the identification of de novo *NR4A2* variants among patients with intellectual disabilities and speech/motor impairments reinforces their significance in 2q23q24 microdeletion syndrome [21] and further spotlights the *NR4A2* gene as a potential target for genetic testing in patients or families with autistic spectrum disorders (ASD). Moreover, understanding the functional consequences of these variants opens new avenues for targeted therapeutic interventions that may involve the modulation of *NR4A2* activity in affected individuals. To date, 43 pathogenic *NR4A2* variants have been described and are strongly associated with different neurodevelopmental disorders [22]. Herein, we aimed to assess the role of *NR4A2* variants in the genetic underpinnings of ASD among Saudi children. The study cohort encompassed different age ranges and included both sexes; however, the male/female ratio of patients (~2.6:1) discords with the global scenario of a male-biased distribution (4:1) among individuals with ASD [23], but it partly agrees with the recently reported ratio of 3:1 in autistic children in Saudi Arabia [20]. On the other hand, our findings might approach the “X chromosome-related protective effect”, in which autistic females have a higher liability threshold, i.e., require a higher threshold of genetic and epigenetic insults for ASD manifestation than males do, despite having the same risk [24]. This knowledge provides a plausible interpretation for the general trend of higher IQ scores observed in female patients than in their male counterparts. Similarly, younger ASD patients presented a similar trend, as indicated by significantly higher scores on the full-scale, verbal, and non-verbal IQ scales, which could be expected given that IQ functioning tends to be relatively stable in children with ASD > 8 years of age [25]. Moreover, the recent findings of Al-Mamari et al. [26] added more credence to our results, considering that their report identified the intellectual profile of autistic children in a relevant regional setting (Oman).

Consistent with previous studies that described the core features of ASD and common comorbidities in affected individuals [3], the autistic children in our cohort displayed varying degrees of neurodevelopmental disorders reflected by impairments in motor, communication, and language skills, and were generally associated with intellectual deficiency. Prominently, gastrointestinal (GI) problems were the most commonly reported phenotype among our cohort, which could be attributable to the disrupted gut microbiome recently highlighted as a key contributor to the development of ASD [27]. Moreover, the impact of GI distress has been reported to be associated with the sleeping abnormalities, developmental delay, and behavioral problems often co-occurring with autism [28], which could explain the high and convergent frequencies of these comorbidities in our patients. In another context, the gut microbiota plays multifaceted key roles in immune homeostasis [29], prominently through the overproduction of IL-12, the central cytokine bridging the innate and adaptive immune systems [30]. Accordingly, it can be expected that disrupted gut microflora is often coherent with a deregulated immune system that consequently triggers several autoimmune disorders, including those observed in our ASD subjects.

Results of the genetic analysis revealed that 2.37% of our ASD cohort carried de novo loss-of-function *NR4A2* variants, a notable frequency considering the genetic heterogeneity often reported in ASD populations [21]. However, this could be attributed to the uniform and comprehensive sequence coverage and more efficient identification of gene-disruptive mutations provided by exome sequencing. Notably, three of the ten identified variants (30%) were recurrent, particularly the nonsense variant c.44_45insA and the CNV (c.1159-81_1540+67del), which co-occurred in five out of eight probands, followed by the c.1del nonsense variant, which was identified in three patients (one proband and two unrelated affected siblings). These findings indicate that recurrent *NR4A2* variants, despite their rarity, might sporadically reoccur in unrelated ASD patients, which conforms to several studies reporting this phenomenon, even in asymptomatic carriers [31]. Furthermore, Wirth et al. [11] reported a recurrence of the nonsense *NR4A2* variant c.326dupA that was previously reported by Ramos et al. [10], taking into consideration that patients reported in both studies were unrelated.

Although de novo *NR4A2* variants were identified in both genders in this study, the prevalence was higher in males (3.28%) than in females (2.13%) in the total cohort, with no significant difference observed in the variant distribution. This male skew was recently ascribed to the sex-differentially expressed genes (DEGs) involved in modulating ASD risk pathways, where they do not preferentially overlap with ASD-associated variants during prenatal cortical development, suggesting that those genes may contribute to ASD risk in other brain regions or cell types, or at other developmental phases [32]. The identified variants were predominantly located within the N-terminal domain (NTD) of the DNA-binding motif (DBM) and the ligand-binding domain (LBD) of the protein, both of which are crucial for *NR4A2* regulatory functions [16]. Considering that the alteration of cell-autonomous dopaminergic functions is the core hallmark of ASD pathophysiology, *NR4A2* mutations have been associated with various disorders related to dopaminergic dysfunction, which contributes to several behavioral manifestations of autism [33]. Therefore, the disruption of these functional domains likely impairs the normal function of *NR4A2*, contributing to the developmental and behavioral phenotypes observed in affected carriers.

Strikingly, deleterious *NR4A2* variants were more frequently identified among multiplex families than among simplex families (8.7% and 2.05%, respectively), arguing against the previous findings of Leppa et al. [34] and Ruzzo et al. [35], who reported that rare de novo protein-truncating mutations are more commonly observed among simplex families, which provides more evidence of the complexity of genetic architecture differences between simplex and multiplex families. However, the high heritability of ASD and the high recurrence risk among ASD siblings compared with the population prevalence (20.2%) support our findings [6,36]. Despite the recurrence of certain pathogenic *NR4A2* variants in quad siblings, their distribution patterns among affected sibling pairs were quite different. In the first quad, both concordant siblings did not share the same disease-causing variants, and even the discordant sibling carried a different VUS variant, which might infer germline mosaicism in one or both parents [37], given that neither of these variants was observed in either parent. Meanwhile, both concordant siblings in the second quad and the discordant sibling carried the same nonsense variant (c.44_45insA) that was inherited from a healthy mother, which co-occurred with other de novo variants in these siblings. This observation was reminiscent of the previous hypothesis of Ye et al. [38], who postulated that the X chromosome-mediated resistance of mothers could enable them to carry pathogenic variants without being affected. It should be considered, though, that the precise genetic landscape of ASD remains intricate and enigmatic, particularly the transmission of deleterious variants from an unaffected parent to their offspring. One possible explanation for this phenomenon is related to the variable penetrance and/or expressivity of ASD risk alleles, which might depend on the combined effects of other genetic and/or environmental triggers [39]. In other words, multiple co-occurring gene-disruptive mutations can additively or synergistically contribute to phenotype manifestation or severity. Additionally, differential transmission of risk alleles combined with the occurrence of de novo variants are frequently observed in multiplex families [39], which together justifies the phenotypic heterogeneity in concordant sibling pairs, even when Mendelian inheritance is involved. This proposition was further consolidated with several reports that accentuated the significant contribution of parental age, for example, to ASD risk in offspring, where de novo mutations accumulate in their germline with age [40]. Similarly, other reports highlighted the epigenetic influence of low-grade inflammation induced by maternal hyperinsulinemia, obesity, and disrupted gut microbiome on the expression of *NR4A2* during prenatal/early-life exposome of offspring [41,42].

The clinical phenotypes observed among affected *NR4A2* subjects followed the general pattern observed in the total cohort; however, we noticed a differential manifestation of ASD symptomatology among the probands from simplex families who carried the same variants and even among affected sibling pairs. These observations are consistent with the phenotypic heterogeneity inherent in ASD symptomatology and further align with the consensus of the absence of genotype–phenotype correlations in *NR4A2*-related phenotypes [10]. Moreover, phenotype severity was more prominent in the female proband of the first quad (A12) than in her affected male sibling, which could be expected considering that autistic females were reported to have a greater burden of protein-truncating mutations and stronger DNA methylation signatures in brain-relevant genes during the prenatal period than males [43,44]. The frequency of language and motor impairments was prominent among the affected *NR4A2* subjects, which could be related to deficient *NR4A2* expression in the brain, particularly in the superior temporal sulcus, which is reportedly implicated in language development and social perception [45], and in the substantia nigra, which is responsible for the production of dopamine, which modulates voluntary movement and muscle tone [46]. In the same vein, the high incidence of GI disorders could be attributed to dysregulated *NR4A2* expression in immune cells [47], where *NR4A2* modulates the production of proinflammatory (Th1) cytokines from CD4^+^ T cells under pathological conditions by trans-activating Foxp3 expression on regulatory T cells (Tregs), i.e., the key regulator of the differentiation and function of Tregs. Therefore, *NR4A2* deletion/dysfunction attenuates the recruitment of Tregs, which exacerbates the production of the proinflammatory cytokines associated with gastrointestinal inflammation [48]. Psychobehavioral issues, including hyperactivity, anxiety, and sleeping problems, have also been reported among affected carriers, mirroring another aspect of the dysregulated dopaminergic system ascribed to ADHD traits and altered locomotor activity [49].

On the other hand, two patients (20%) had relatively high FSIQ scores (>70), suggesting that protein-truncating mutations play a less prominent role in ASD symptomatology in high-functioning patients [43]. The small sample size of the *NR4A2* cohort limited the analysis of the distribution of different IQ subscale scores across different age ranges or sexes; nevertheless, the younger and female *NR4A2* subjects tended to have overall higher FSIQ, NVIQ, and Vineland scores than their male counterparts did, following the general pattern observed in the total cohort. Considering that no other potentially pathogenic variants in genes associated with Rolandic epilepsy or intellectual deficiency were detected in our *NR4A2* cohort, the absence of the identified variants in large population databases (gnomAD and ExAC) and the lack of prior clinical significance assessments in ClinVar suggest their potential pathogenicity. While *NR4A2* is considered a monogenic disease-causing gene [10,11,12,13], our findings conform to the broader perception of de novo variants, particularly those causing loss of function, as significant contributors to ASD risk.

Of interest, the ES analysis revealed that five (35.72%) discordant siblings carried potentially pathogenic *NR4A2* variants, two (40%) of which had SNVs/indels that were absent in their affected siblings. Considering that most of the potentially pathogenic *NR4A2* variants identified in discordant siblings occurred de novo, we assumed that the absence of phenotypes is related to differential penetrance and/or expressivity of these variants, whereas the multiple protein-truncating mutations observed in the affected carriers might have additively contributed to the induction of ASD manifestations [39]. Regarding the maternally inherited c.44_45insA nonsense variant recognized in the discordant sibling of the second quad, we hypothesized that there might be two types of shared *NR4A2* variants, a “strong” variant and a “weak” variant, as previously proposed by Ye et al. [38]. Maternal factors (e.g., obesity, chronic inflammatory diseases), paternal age, and prenatal stress may influence the penetrance and expressivity of these variants [41,42].

According to our PCR analysis results, the CNV was predicted to be a LOF variant affecting canonical splicing and leading to exon loss; therefore, it is likely the “strong” variant associated with the clinical phenotypes observed in the affected siblings was absent in the unaffected siblings. Meanwhile, c.44_45insA is a weak variant that, when present without the CNV, does not induce a clinical phenotype. Additionally, the X chromosome could provide the more protective effects associated with the absence of clinical phenotypes despite carrying potentially deleterious *NR4A2* variants [24], given that 80% of the discordant carriers are females. It should be considered, however, that while the onset often occurs in early childhood, ASD symptomatology may not fully manifest until later, when social demands exceed limited capacity [50]; therefore, the follow-up of discordant carriers is recommended. To elucidate the broader molecular impact of NR4A2 variants, we recommend large-scale multicenter genomic studies supplemented with proteomic and metabolomic profiling. These approaches could help identify downstream pathways and potential biomarkers for therapeutic targeting.

## 4. Materials and Methods

### 4.1. Study Cohort

The participants in this study included consecutive pediatric patients who were referred to the Center for Autism Research Department of King Faisal Specialist Hospital & Research Centre (Riyadh-KSA) between 2019 and 2023. The assessment of ASD was conducted by a multidisciplinary team that included board-certified developmental pediatricians who assessed autistic traits based on the Diagnostic and Statistical Manual of Mental Disorders (DSM-5) criteria [51] and speech therapists, psychologists, and social workers. Intellectual functioning was assessed using the Stanford–Binet Intelligence Scales, Fifth Edition (SB5) [52], providing FSIQ (full scale IQ), VIQ (verbal IQ), and NVIQ (non-verbal IQ) scores. However, additional standardized scales were included to ensure diagnostic accuracy and assess symptom severity. These scales included the social responsiveness scale (SRS) that evaluates the severity of ASD symptoms across social awareness, cognition, communication, motivation, and mannerisms [53], the Peabody Picture Vocabulary Test (PPVT) for assessment of receptive vocabulary development [54], and Vineland Adaptive Behavior Scales that measure the adaptive functioning, including communication, daily living skills, socialization, and motor skills [55]. According to the obtained intellectual information, six cognitive profile groups were considered in this study: average and above IQ (90–129), low average IQ (80–89), borderline (IQ = 70–79), mild impairment (IQ = 55–69), moderate impairment (IQ = 40–54), and severe impairment (IQ < 35) [26].

### 4.2. Study Design

Clinical and demographic data were collected from the medical records of children with ASD admitted to the Centre of Excellence in Genomic Medicine Research (CEGMR) at King Abdul-Aziz University (Jeddah-KSA). The inclusion criteria included confirmed ASD diagnosis based on DSM-5 criteria, availability of both parents for trio analysis, and being aged between 3 and 14 years. Exclusion criteria included the presence of unrelated somatic or chronic disorders (e.g., atopic allergies, autoimmune diseases, or metabolic syndromes) that might confound the clinical phenotype. Ethical approval for the study was obtained from the Local Ethical Committee of King Abdul-Aziz University (reference number: HA-02-J-003).

Samples of peripheral blood samples (5–8 mL) were obtained from all participants in EDTA-coated tubes. Peripheral blood mononuclear cells (PBMCs) were isolated within 4 h using Ficoll^®^400 (Sigma Aldrich, St. Louis, MO, USA) density gradient centrifugation and stored in RNAlater (Thermo Fisher Scientific, Waltham, MA, USA) at −80 °C until DNA extraction. The identification of *NR4A2* variants was conducted in two phases: a screening phase, in which 338 children with ASD from 315 unrelated families (293 simplex, 2 quads, and 1 quintet) who met the criteria were subjected to exome sequencing (ES), and a second phase, in which patients with identified *NR4A2* variants were subjected to trio genetic analysis along with their parents, affected siblings (if any), and unaffected siblings.

### 4.3. Genetic Analysis and Variant Validation

The genomic DNA required for exome/Sanger sequencing was extracted from the PBMCs using a QIAamp DNA Blood Mini Kit (QIAGEN, Hilden, Germany) according to the manufacturer’s instructions. For exome sequencing (ES), IDT xGen Exome Research Panel v1.0 (IDT, Coralville, IA, USA) was used to capture exons for subsequent analysis on a HiSeq X10 platform (Illumina, San Diego, CA, USA) with ≥100 bp paired-end reads. Reads were mapped to the human genome assembly GRCh38, and the sequence variation was analyzed via FASTQ data according to Xiaozhen et al. [56]. All candidate variants were validated by Sanger sequencing, and the de novo status was confirmed by either ES or Sanger sequencing of the parental genome. The primers used for complementary DNA (cDNA) synthesis, PCR amplification, and Sanger sequencing were designed via Primer3 (version 0.4.0) to span 50 bp upstream and downstream of each exon to include putative variants in splicing regions.

### 4.4. Functional Analysis of Splice Region Variants in ASD Patients with Chromosomal Deletions

Patients with genomic deletions were subjected to targeted PCR amplification of the region flanking the putative variant to examine its potential functional consequences on canonical splicing. RNA was extracted from the PBMCs using RNeasy kit (Thermo Fisher Scientific, USA) according to the manufacturer’s instructions and then used for cDNA synthesis using iScript^TM^ RT Supermix (Bio-Rad, Hercules, CA, USA). The reaction mixture included 1 µg of purified RNA, 4 μL of 5X iScript^TM^ Supermix, and each of the following primers at a final concentration of 10 µM: *F*: 5′-CTATGACCAGCCTGGACTATTC-3′ and *R*: 5′-TCCCCAACAGTTTGGACAAAT-3′. The final reaction volume was 20 μL. The thermal cycling program included a 5-min round of priming at 25 °C, followed by reverse transcription for 20 min at 46 °C and a final round of enzyme inactivation at 95 °C for 1 min. The cDNA was further used as a template for PCR using the same primer set to confirm altered splicing. The reaction mixture included 100 ng of cDNA, 10 μL of 5X Phusion HF Buffer, 1 μL of dNTPs mixture (10 mM each) (Thermo Fisher Scientific, USA), 0.5 µM of each primer (final concentration), 1U of Phusion™ Hot Start II High-Fidelity DNA Polymerase (Thermo Fisher Scientific, Waltham, MA, USA), and 1.5 μL of DMSO, and the reaction volume was 50 μL with nuclease-free H_2_O. The cycling conditions included one pre-denaturation cycle at 98 °C for 30 s, followed by 35 cycles of 10 s at 98 °C for denaturation, 20 s at 56 °C for annealing, 30 s at 72 °C for extension, and a final extension cycle for 5 min at 72 °C. The amplicons were visualized in 1% agarose premixed with 0.3% ethidium bromide.

### 4.5. Variant Annotation and Classification

Variants were annotated based on the *NR4A2* transcript NM_006186.4, and their pathogenicity was classified according to the guidelines of the American College of Medical Genetics and Genomics and the Association for Molecular Pathology (ACMG/AMP) [57] and ClinGen specifications [58]. Variant description was confirmed via the Mutalyzer (2.0.35) tool, whereas altered splicing and potential loss of function (LoF) were predicted via several in silico tools, including SpliceAI Lookup, NNSPLICE, and MaxEntScan tools, and the missense variants were evaluated via MutationTaster, CADD, SIFT, and Polyphen-2 [12].

### 4.6. Statistical Analysis

The sample size was calculated via a single proportion formula based on the previous report of Eapen et al. [59], which had a calculated sample size of 317. The statistical power of our sample size was further assessed via the Priori test in G*Power software (v3.1.9.7), with an admissible range between −20 and 20 for the effect size (d). The analysis of two independent groups included in the study revealed a 1.427-fold difference in the FSIQ score in the ASD group relative to the discordant group (unaffected siblings), with average sample sizes of 17 and 7 for the ASD and discordant sibling groups, respectively, which were required to achieve an effect size (d) of 1.427 and a study power of 95% (1-β error probe) (Appendix A). A continuity-corrected squared Fisher’s exact test was used to evaluate the null hypothesis with a probability of type I error (α-error = 0.05), power = 95%. Statistical analyses were performed using GraphPad Prism 9. Data normality was assessed using the Shapiro–Wilk test. For group comparisons, one-way ANOVA followed by Tukey’s post hoc tests was applied for parametric data. Non-parametric data were analyzed using Kruskal–Wallis tests followed by Dunn’s post hoc corrections. For adaptive behavior evaluations using age-equivalent Vineland scores, the ASD cohort was stratified using a mean split of 4.5 years, as per Grondhuis et al. [25]. The frequency distribution was also used to describe the demographic, clinical, and IQ scores of the children with ASD. *p* values ≤ 0.05 were considered statistically significant.

## 5. Conclusions

The current report provides additional evidence regarding the role of the *NR4A2* gene in autism spectrum disorder (ASD) among children in Saudi Arabia. These findings contribute to the growing body of knowledge on the genetic landscape of autism and highlight the need for more investigations to evaluate the potential of *NR4A2* as a candidate therapeutic target in ASD patients. The high frequency of recurrent variants reported in our cohort signifies their contribution to ASD etiopathogenesis and suggests their eligibility as diagnostic candidates for genetic counseling and early intervention, particularly in patients with intellectual disability and language impairments. Additionally, the finding that some unaffected siblings shared/carried deleterious *NR4A2* variants underscores the necessity of further investigations to reveal the genetic and environmental factors that modulate the penetrance and expressivity of these variants. 

## 6. Limitations of the Study

While the sample size was relatively large for a single-center study, it was one of the limiting factors in this report, particularly regarding the number of *NR4A2* subjects, which limited the subgroup analysis of the age/gender distribution of intellectual functioning scores or potential genotype–phenotype correlations. Although our findings were derived from a certain ethnic group, the multiple occurrences of recurrent variants are worthy of further investigation to assess their generalizability to other ethnicities. The relatively high frequency of de novo *NR4A2* variants in our cohort may be influenced by the nature of our referral center, which may attract more complex or syndromic ASD cases. Ongoing investigations are focused on replicating these results within a larger and more diverse population and exploring their potential interactions with other genetic and environmental factors, aiming to reach a conclusive understanding of the precise role of these variants in the etiopathogenesis of autistic children. Future work employing quantitative RNA sequence analysis or long-read transcriptome sequencing is also warranted to further characterize the nature and consequence of the splicing disruption

## Figures and Tables

**Figure 1 ijms-26-05468-f001:**
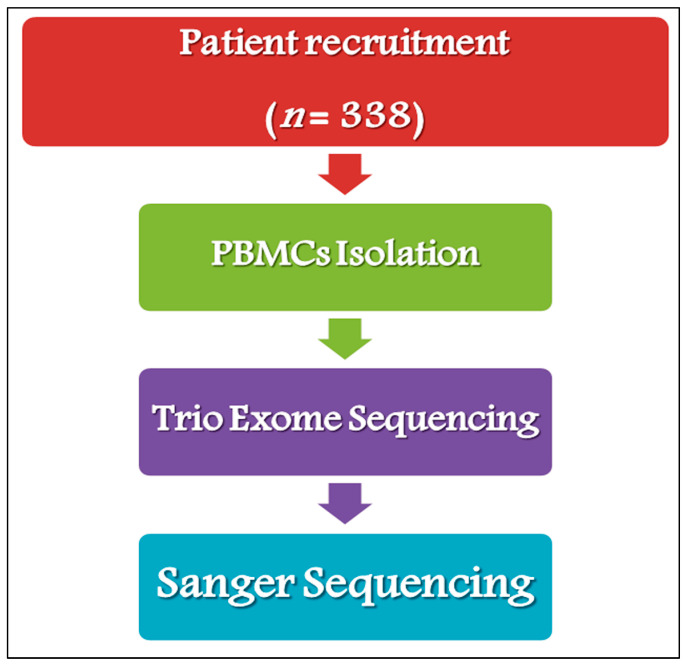
Overview of the genetic analysis workflow for Saudi children with ASD. The molecular screening for *NR4A2* variants was performed in two stages. In the first stage, 338 children with ASD from 315 unrelated families who met our inclusion criteria were subjected to exome sequencing (ES) using PBMC-derived genomic DNA. The second stage included a trio analysis for probands with identified *NR4A2* variants, followed by Sanger sequencing for variant validation.

**Figure 2 ijms-26-05468-f002:**
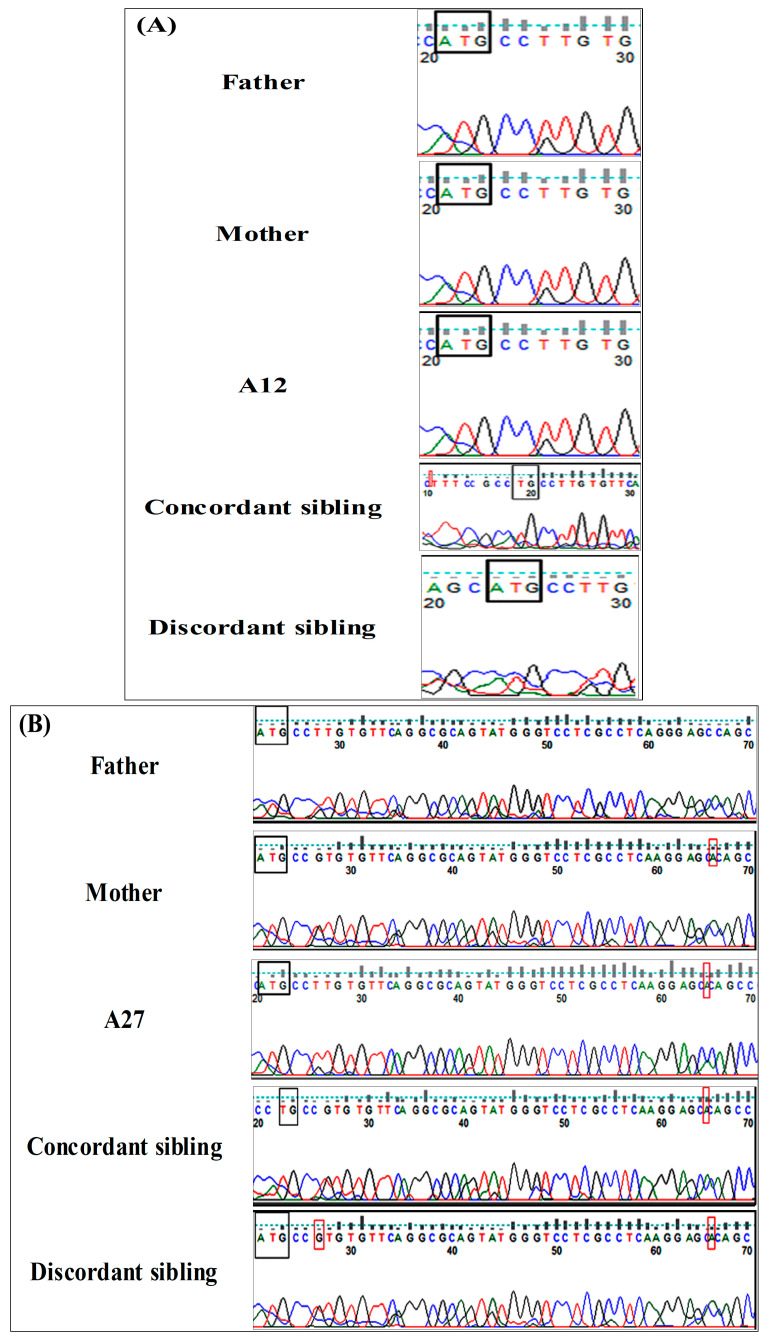
Genetic analysis of probands identified with *NR4A2* variants. As revealed by trio exome/Sanger sequencing, the two affected siblings of the first quad (**A**) had different pathogenic variants, where the proband (A12) carried the recurrent CNV, a missense variant (c.5_6delCTinsTG, p.P2L), and a nonsense variant (c.534del, p.F178*), whereas the affected sibling (AS_12) carried an intronic indel (c.-2-8del) located in intron 2 and the c.1del nonsense variant of the NTD. Similarly, the unaffected sibling (B12) carried a different de novo missense variant (c.536_537delCTinsGC, p.K179S), and neither of the variants was identified in the parents’ genomes. Regarding the second quad (**B**), both affected siblings and the unaffected sibling shared the nonsense variant c.44_45insA (p.S16*) that was identified in the healthy mother’s genome. However, the proband (A27) also carried a pathogenic CNV and another VUS missense variant (c.537G>C), while the nonsense variant c.1del was also identified in the affected sibling (AS_27). All the co-occurring variants identified in both affected siblings were de novo and were absent in their unaffected siblings. Another two discordant siblings (B3_2 and B11_2) of two simplex families carried potentially pathogenic de novo insertions (c.30_31insG, p.S11* and c.14_15insT, p.Q5*, respectively) that were absent in their probands (**C**,**D**). Black squares determine the start codon, and the red squares represent the nucleotide insertion/deletions.

**Figure 3 ijms-26-05468-f003:**
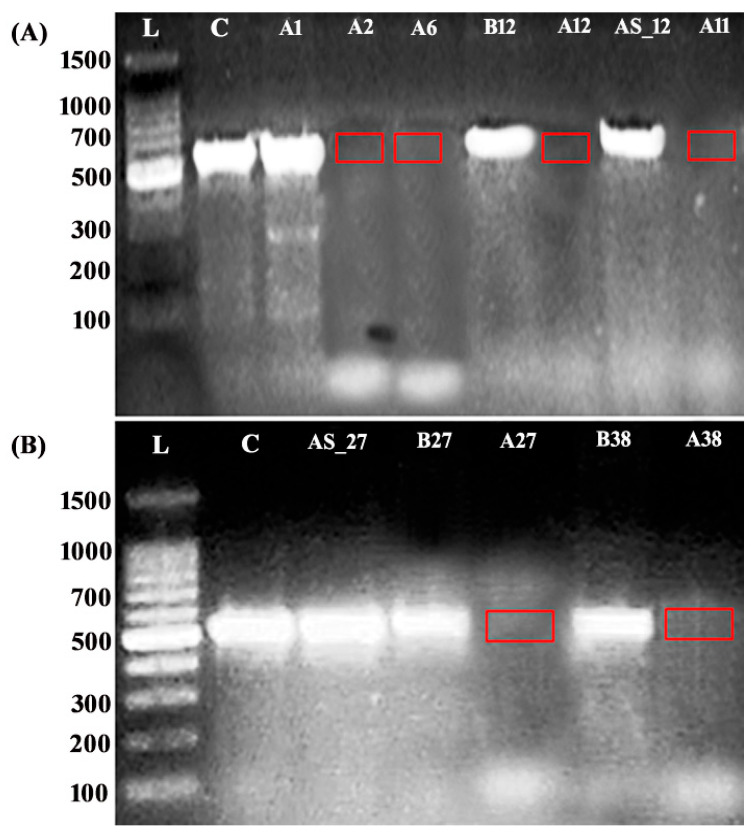
Functional analysis of the splice region variant (CNV). (**A**,**B**): Patients with chromosomal deletions were subjected to targeted RT–PCR followed by PCR amplification of the region flanking the putative variant to examine its potential functional consequences on canonical splicing. The results revealed aberrant transcripts that were undetectable at the expected size (~540 bp) in the amplicons obtained from probands carrying the CNV (red squares) compared with those obtained from both concordant and discordant siblings, as well as a control proband (A1) who did not carry this variant (**A**), which confirmed the loss of exons 6 and 7.

**Figure 4 ijms-26-05468-f004:**
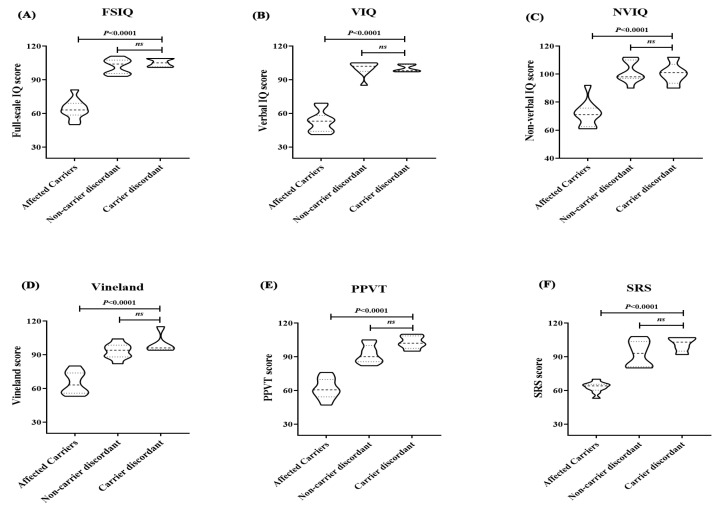
Violin plots represent the distribution and density of the intellectual and behavioral scores (FSIQ, VIQ, NVIQ, Vineland, PPVT, and SRS) among affected and unaffected *NR4A2* variant carriers. The mean scores of FSIQ (**A**), as well as the verbal and non-verbal IQ (**B**,**C**), Vineland (**D**), PPVT (**E**), and SRS (**F**) subscales, were significantly lower in affected carriers than in both carrier and non-carrier discordant siblings (*p* < 0.0001/for all), with no significant difference observed between the latter groups. The thick central line denotes the median, and the thinner lines indicate the inter-quartile range. Statistical comparisons were performed using one-way ANOVA followed by Tukey’s multiple comparisons test. Ns: non-significant.

**Table 1 ijms-26-05468-t001:** General demographic and clinical profile of the study cohort (n = 338).

Variable	
Gender [n, %]	
Male	244 (72.19%)
Female	94 (27.81%)
Age (Years) [n, %]	
3–5	74 (21.89%)
6–9	125 (36.98%)
10–14	139 (41.13%)
Family history	
Simplex	293 (93.01%)
Quad	21 (6.66%)
Quintet	1 (0.317%)
Total	315 (100%)
FSIQ classification [n, %]	
Average and above IQ (90–129)	33 (9.76%)
Low average IQ (80–89)	59 (17.46%)
Borderline IQ (70–79)	74 (21.89%)
Mildly impaired IQ (55–69)	84 (24.85%)
Moderately impaired IQ (40–54)	72 (21.3%)
Severely impaired IQ (>35)	16 (4.74%)
IQ subscales scoring	
FSIQ score	84 (68.5–100.5) *
VIQ score	81.5 (63–101.5)
NVIQ score	91 (75–104)
PPVT score	77 (67.5–88.5)
SRS score	77 (67–90)
Vineland score (AE)	
<4.5 Years	102 (97–112)
>4.5 Years	69 (57–83)
Clinical Phenotypes (current) [n, %]	
Gastrointestinal disorders	182 (76.47%)
Sleeping disorders	165 (48.82%)
Language/speech impairment	160 (47.34%)
Motor difficulties	148 (43.79%)
Feeding difficulties	136 (40.23%)
Short stature	77 (22.78%)
Seizures/epilepsy	35 (10.35%)
Autoimmune diseases	33 (9.76%)
Facial dimorphism	17 (5.03%)
Psychiatric/behavioral problems (current) [n, %]	
Hyperactivity/ADHD	185 (54.74%)
Anxiety	143 (42.3%)
Sensory processing difficulties	123 (36.39%)
Learning difficulties	115 (34.02%)
Attachment disorder	97 (28.7%)
OCD	33 (13.87%)
Aggressive behavior	15 (4.43%)

Abbreviations: IQ: intelligence quotient; FSIQ: full scale IQ; VIQ: verbal IQ; NVIQ: non-verbal IQ; PPVT: Peabody picture vocabulary test standardized score; SRS: social responsiveness scale; AE: age equivalent; OCD: obsessive–compulsive disorder; and ADHD: attention deficit hyperactivity disorder. * All values are represented as median (25th–75th percentiles).

**Table 2 ijms-26-05468-t002:** De novo *NR4A2* variants identified in children with ASD.

Proband (Biosample Accession)	Age (Y)/Sex	Variant (NM_006186.4)	Clingen Identifier/ClinVar Accession	Protein Alteration	SNV/Indel/ CNV	Protein Domain/Region	ACMG Classification/Evidences	Clinical Phenotypes/Behavioral Problems	Seizures?
A1 (SAMN43045798)	6/M	c.-2-2del	VCV003338047.1	Splicing change	Splice region variant	Intron 2	Pathogenic (12 ACMG Points: 12P and 0B. PVS, PM2, PP3 (moderate))	Mild ID, absent speech, mild ataxia, stereotypical body rocking, irregular sleeping, social phobia, attachment disorder, hypersensitivity to external stimuli	No
		c:1del	CA2837995541/VCV003338025.1	p.(Met1del)	Indel	NTD	Pathogenic (10 ACMG Points: 10P and 0B. PVS1, PM2)		
		c.548del	CA645529278, VCV003338477.1	p.(Pro183Leufs*20)					
A2 (SAMN43045230)	11/M	c.44_45insA	CA2837995542, VCV003338013.1	p.(Ser16Glnfs*28)	Indel	NTD	Pathogenic (10 ACMG Points: 10P and 0B. PVS1, PM2)	Moderate ID, significant language impairment, Rolandic epilepsy, feeding problems, sleeping problems, choreoathetotic movements, incontinence, hyperactivity, learning difficulties	Yes
		c.1159-81_1540+67del	CA2838010100, VCV003338024.1	p.(Phe387Argfs*19)/Splicing change	CNV	LBD			
A3 (SAMN43045749)	11/M	c.44_45insA	CA2837995542, VCV003338013.1	p.(Ser16Glnfs*28)	Indel	NTD	Pathogenic (10 ACMG Points: 10P and 0B. PVS1, PM2)	Mild ID, delayed speech and motor development, mild ataxia, dystonia, chronic dyspepsia and constipation, hyperactivity, anxiety	No
		c.536del	CA2837995544, VCV003338030.1	p.(Lys179Serfs*24)					
A6 (SAMN43046049)	4/F	c.44_45insA	CA2837995542, VCV003338013.1	p.(Ser16Glnfs*28)	Indel	NTD	Pathogenic (10 ACMG Points: 10P and 0B. PVS1, PM2)	Mild ID, speech dyslexia, delayed walking, mild hypotonia, chronic diarrhea, hyposensitivity to temperature and pain, attachment disorders	No
		c.1159-81_1540+67del	CA2838010100, VCV003338024.1	p.(Phe387Argfs*19)/Splicing change	CNV	LBD			
A11 (SAMN43045260)	7/M	c.44_45insA	CA2837995542, VCV003338013.1	p.(Ser16Glnfs*28)	indel	NTD	Pathogenic (10 ACMG Points: 10P and 0B. PVS1, PM2)	Mild to moderate ID, speech and motor delay, learning difficulties, generalized hypotonia, frequent diarrhea, hypermobile EDS, skin hyperextensibility, mild scoliosis, sleeping problems	Infantile spasms
		c.1159-81_1540+67del	CA2838010100, VCV003338024.1	p.(Phe387Argfs*19)/Splicing change	CNV	LBD			
A12 (SAMN43046050)	10/F	c.5_6delCTinsTG	CA2837995543, VCV003338026.1	p.(Pro2Leu)	SNV	NTD	VUS (3 ACMG points: 3P and 0B. PM2, PP3). SIFT score = 0 (deleterious), PolyPhen score = 0.929 (probably damaging)	Moderate ID, prominent speech impairment, supported walking, overall poor growth, progressive hypotonia, short stature, facial dimorphism, IBD, sleeping disorders	Lennox-Gastaut syndrome
		c.534del	CA2837995545, VCV003338016.1	p.(Phe178Leufs*25)	Indel		Pathogenic (10 ACMG Points: 10P and 0B. PVS1, PM2)		
		c.1159-81_1540+67del	CA2838010100, VCV003338024.1	p.(Phe387Argfs*19)/Splicing change	CNV	LBD			
AS_12 (SAMN43045274)	9/M	c.-2-8del	VCV003338474.1	Non coding	Splice polypyrimidine tract variant	Intron 2	VUS/Likely Benign (-1 ACMG points: 1P and 2B. BP4, PM2)	Mild ID, speech apraxia, delayed echolalia, choreoathetotic movements, supported walking, IBD, incontinence, anxiety, irregular sleeping	No
		c:1del	CA2837995541/VCV003338025.1	p.(Met1del)	Indel	NTD	Pathogenic (10 ACMG Points: 10P and 0B. PVS1, PM2)		
A27 (SAMN43045261)	8/M	c.44_45insA	CA2837995542, VCV003338013.1	p.(Ser16Glnfs*28)	Indel	NTD	Pathogenic (10 ACMG Points: 10P and 0B. PVS1, PM2)	Mild ID, moderate speech impairment, disproportionate motor development, generalized hypotonia, ataxic gait, anxiety, attachment disorders	No
		c.537G>C	CA348682205, VCV003338475.1	p.(Lys179Asn)	SNV		VUS (2 ACMG points: 2P and 0B. PM2, PP2). TraP score = 0.338		
		c.1159-81_1540+67del	CA2838010100, VCV003338024.1	p.(Phe387Argfs*19)/Splicing change	CNV	LBD	Pathogenic (10 ACMG Points: 10P and 0B. PVS1, PM2)		
AS_27 (SAMN43046045)	5/M	c:1del	CA2837995541/VCV003338025.1	p.(Met1del)	Indel	NTD	Pathogenic (10 ACMG Points: 10P and 0B. PVS1, PM2)	Mild ID, receptive-expressive language delay, motor delay, mild infantile hypotonia, motor tics, IBD, anxiety, hyperactivity	No
		c.44_45insA	CA2837995542, VCV003338013.1	p.(Ser16Glnfs*28)					
A38 (SAMN43045262)	10/M	c.39A>G	CA429727752	p.(Gln13=)	Synonymous	NTD	Likely benign (3 ACMG points: 2P and 5B. PM2, PB4, BP7)	Mild to moderate ID, significant language impairment, ataxic gait, dystonia, sleeping disorders, self injury, aggressive behavior	Febrile
		c.44_45insA	CA2837995542, VCV003338013.1	p.(Ser16Glnfs*28)	Indel		Pathogenic (10 ACMG Points: 10P and 0B. PVS1, PM2)		
		c.1159-81_1540+67del	CA2838010100, VCV003338024.1	p.(Phe387Argfs*19)/Splicing change	CNV	LBD			

Abbreviations: M: male; F: female; SNV: single nucleotide variant; indel: insertion/deletion; CNV: copy number variant; NTD: N-terminus domain; LBD: ligand-binding domain; ACMG: American College of Medical Genetics and Genomics; and VUS: variant of uncertain significance. * All probands carry de novo cis-heterozygous *NR4A2* variants.

**Table 3 ijms-26-05468-t003:** Frequency of de novo *NR4A2* variants in discordant siblings of ASD probands.

Patient (Biosample Accession)	Age (Y)/Sex	Variant (NM_006186.4)	Clingen Identifier/ClinVar Accession	Protein Alteration	SNV/Indel	Protein Domain/Region	ACMG Classification	ACMG Evidences
B3_2 (SAMN43942449)	6/F	c.30_31insG	CA2838140773/ VCV003338478.1	p.(Ser11Valfs*33)	Indel	NTD	Pathogenic	10 ACMG Points: 10P and 0B. PVS1, PM2 (moderate)
B11_2 (SAMN43942450)	3/F	c.14_15insT	CA2838140772/ VCV003338476.1	p.(Gln5Hisfs*39)	Indel		Pathogenic	10 ACMG Points: 10P and 0B. PVS1, PM2 (moderate)
B12 (SAMN43942448)	7/M	c.536_537delCTinsGC	PA2838140774/VCV003338031.1	p.(Lys179Ser)	SNV		VUS	2 ACMG points: 2P and 0B. PM2 (moderate)
B27 (SAMN43370295)	4/F	c.5_6delCTinsTG	CA2837995543, VCV003338026.1	p.(Pro2Leu)	SNV		VUS	3 ACMG points: 3P and 0B. PM2, PP3). SIFT score = 0 (deleterious), PolyPhen score = 0.929 (probably damaging)
		c.44_45insA	CA2837995542/VCV003338013.1	p.(Ser16Glnfs*28)	Indel		Pathogenic	10 ACMG Points: 10P and 0B. PVS1, PM2 (moderate)
B38 (SAMN43370296)	6/F	c.39A>G	CA429727752	p.(Gln13=)	Synonymous		Likely benign	3 ACMG points: 2P and 5B. PM2 (Supporting), PB4 (strong), BP7

Abbreviations: M: male; F: female; SNV: single nucleotide variant; indel: insertion/deletion; NTD: N-terminal domain; ACMG: American College of Medical Genetics and Genomics; and VUS: variant of uncertain significance. * All subjects carry de novo cis-heterozygous *NR4A2* variants.

## Data Availability

Inquiries about data availability should be directed to the corresponding author upon reasonable request owing to the privacy of patients’ results.

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
