# Peer review of "Molecular Screening Reveals De Novo Loss-of-Function NR4A2 Variants in Saudi Children with Autism Spectrum Disorders: A Single-Center Study"

_ijms, 2025, doi:10.3390/ijms26125468_

Round 1

Reviewer 1 Report

Comments and Suggestions for Authors

The manuscript written by Alharbi et al, titled “Molecular screening reveals de novo Loss-of-function NR4A2 variants in Saudi children with Autism Spectrum Disorders,” reports a detailed molecular identification of variants of the gene encoding the nuclear receptor superfamily 4 group A member 2 (NR4A2) present in a population of children with autism spectrum disorders (ASD), also reporting their association with pathogenic features of loss of function of the translated protein.

Relatively few studies have observed the impact on the expression of ASD phenotypes related to the presence of NR4A2 gene variants, indels, nonsense, missense, or involving splicing regions, or even other types of mutations. The presence of these types variants has already been reported in previous work as very rare and penetrant ones. This investigation found a relatively higher than previously reported frequency of these variants in about 340 Saudi Arabian children with ASD from 315 unrelated families, also detecting de novo ones linked to loss-of-function of the encoded protein; interestingly three recurrent variants were found among both affected and unaffected carriers. This study is therefore relevant in the global genetic landscape of ASD. The article is well presented and well written. I have only some few points that require revision by the authors to make the manuscript even better as a whole.  

Abstract

Lines 38 and 39: The authors should change this sentence as follows: “ Three NR4A2 variants were notably recurrent among both affected and unaffected carriers.”

Line 43: The authors should modify the term “etiology” into “etiopathogenesis”. 

Introduction

General comment: human genes should be reported in italic characters: the authors should check also in the whole manuscript eventual typos.

Lines 64-65: This sentence should be improved: “.. which are believed to account for ~ 30% of cases with ID and autism [5] , and the estimates of high heritability reported in numerous family studies suggesting strong genetic contributions to ASD risk,  ..”

Results

Table 1. This Table should be improved in its presentation:

titles within the table should all be in the same format: so, also Family History, Clinical Phenotypes etcc. Should be presented as Gender, Age, FSIQ classification, etcc; I suggest to delete the heading Intellectual functioning, since this is implicit. Also I suggest to change line 164 Clinical Phenotypes (present) with Clinical Phenotypes (current), and Line 174 Psychiatric/behavioral problems (present) with Psychiatric/behavioral problems (current), or. alternatively, specify in the legend. Psychiatric/behavioral problems is presented with an Italic font, please uniform it. In Table 1 caption, the authors should explain : “*All values are represented as medians (min-max) ± SD”. Indeed, usually medians are central values in a dataset usually reported when the variables are not normally distributed, accompanied  by dispersion values as 90 th or 75 th  and 10 th  or 25th  inter-quartiles; SD are usually reported as mean. If this is not a typo, the authors should report in the dedicated paragraph 4.6 Statistical Analysis how they decided to report their data and why.

Figure 3. The authors should better describe the figure: the section A) and the section B), and corresponding lanes.

Figure 4. The authors should detail what represent the violin plots in each graph and the statistical inferential test used, as well as the post-hoc tests used. The same should be described in the 4.6 Statistical Analysis paragraph.

Materials and Methods

4.1 Study Cohort

When informing about the instruments used to appraise the intellectual functioning of patients, even if the authors reported the corresponding citation, it is always recommended to provide a brief description of each evaluation instrument employed, e.g. FSIQ, VIQ, NVIQ etc... This is important to make it immediately clear to readers, even those who are not experts in the field, about the type of evaluation carried out. This can increase the overall impact of the work.

4.2 Study Design

A more detailed description of the selection criteria should be provided. For instance, even if this is a genetic investigation, the authors should report if children presenting somatic disturbances other than those reported in Table 1, for instance atopic allergies or any other clinical condition that could impact the presenting phenotypes. The Approval of the Local Ethical Committee should be also added in this section, even if it is reported more precisely at the end of the manuscript.

4.3 Genetic Analyses and Variant Validation

From “..paired-end reads. Reads were mapped….include putative variants in splicing regions”, the font is in a smaller size than the rest of the text.

Figure 5. The authors should report in the picture, n= instead of n- ; the figure appears somehow unclear: it seems to contain already the results obtained as concerns the presence of the NR4A2 variants. Please try to simplify this graph just reporting the schema of the different steps of the analysis.

The authors should provide a short description of the procedure of blood sampling and PBMC isolation. The same should be done for all other procedures.

4.6 Statistical Analysis

As indicated previously the authors should report the descriptive and inferential statistics used, with a short motivation.

Discussion

The discussion is well presented. Just maybe the authors should provide some hypotheses about the potential role of NR4A2 mutant protein in the ASD phenome. They should also discuss more the gene x environment effects in respect to the 3 variants present in non-affected and affected carriers. They should also encourage at the end of the discussion to prospect further deeper  genomic investigations in a greater number of subjects through multi-center investigations; as well, proteomic and metabolomic analyses of these subjects should be developed, in order to elucidate the etiopathogenetic impact of these variants also at the molecular phenotype level.  

Author Response

First, we would like to express our gratitude to the respected editors and reviewers for their insightful comments on our manuscript. We were able to incorporate changes to reflect the suggestions provided, and we hope that we have given them the maximum consideration. Otherwise, we remain available for any other comments to improve the scientific quality of the manuscript. We have highlighted the changes within the Track Changes Version of the manuscript. Our point-by-point responses to the reviewer’s comments and concerns are provided below.

Reviewer #1

The manuscript written by Alharbi et al, titled “Molecular screening reveals de novo Loss-of-function NR4A2 variants in Saudi children with Autism Spectrum Disorders,” reports a detailed molecular identification of variants of the gene encoding the nuclear receptor superfamily 4 group A member 2 (NR4A2) present in a population of children with autism spectrum disorders (ASD), also reporting their association with pathogenic features of loss of function of the translated protein.

Relatively few studies have observed the impact on the expression of ASD phenotypes related to the presence of NR4A2 gene variants, indels, nonsense, missense, or involving splicing regions, or even other types of mutations. The presence of these types variants has already been reported in previous work as very rare and penetrant ones. This investigation found a relatively higher than previously reported frequency of these variants in about 340 Saudi Arabian children with ASD from 315 unrelated families, also detecting de novo ones linked to loss-of-function of the encoded protein; interestingly three recurrent variants were found among both affected and unaffected carriers. This study is therefore relevant in the global genetic landscape of ASD. The article is well presented and well written. I have only some few points that require revision by the authors to make the manuscript even better as a whole.  

Abstract

  • Lines 38 and 39: The authors should change this sentence as follows: “Three NR4A2variants were notably recurrent among both affected and unaffected carriers.”
  • Response: Thank you for your valuable suggestion. Done. The text is updated accordingly. (Kindly review the Track-changes version (yellow highlighted)).
  • Line 43: The authors should modify the term “etiology” into “etiopathogenesis”.
  • Response: Thank you for your insight. Done (Kindly review the Track-changes version, Abstract, lines 29 and 43; Introduction, line 64; Conclusion, line 673, and Limitations of the Study, line 691 (yellow highlighted)).

Introduction

  • General comment: human genes should be reported in italic characters: the authors should check also in the whole manuscript eventual typos.

  • Response: Thank you for your reminder. Done. NR4A2gene and its variants were modified to italics thorough the manuscript (Kindly review the Track-changes version (cyan highlighted)).

  • Lines 64-65:This sentence should be improved: “.. which are believed to account for ~ 30% of cases with ID and autism [5] , and the estimates of high heritability reported in numerous family studies suggesting strong genetic contributions to ASD risk,  ..”
  • Response: Thank you for your suggestion. Done (Kindly review the Track-changes version, (yellow highlighted)).

Results

  • Table 1.This Table should be improved in its presentation: titles within the table should all be in the same format: so, also Family History, Clinical Phenotypes etcc. Should be presented as Gender, Age, FSIQ classification, etcc; I suggest to delete the heading Intellectual functioning, since this is implicit. Also I suggest to change line 164 Clinical Phenotypes (present) with Clinical Phenotypes (current), and Line 174 Psychiatric/behavioral problems (present) with Psychiatric/behavioral problems (current), or alternatively, specify in the legend. Psychiatric/behavioral problems is presented with an Italic font, please uniform it. In Table 1 caption, the authors should explain: “*All values are represented as medians (min-max) ± SD”. Indeed, usually medians are central values in a dataset usually reported when the variables are not normally distributed, accompanied by dispersion values as 90th or 75 th  and 10 th  or 25th  inter-quartiles; SD are usually reported as mean. If this is not a typo, the authors should report in the dedicated paragraph 4.6 Statistical Analysis how they decided to report their data and why.

  • Response: Thank you for your suggestion. Table 1 was modified accordingly. Regarding the IQ subscales scoring, values were updated to be expressed as median (25th – 75th Percentiles). The table caption was modified accordingly (Kindly review the Track-changes version, yellow highlighted).

  • Figure 3.The authors should better describe the figure: the section A) and the section B), and corresponding lanes.

  • Response: Thank you for your insight. The figure is now updated to include the names of probands carrying the CNV and their concordant/discordant siblings (if any), as well as one proband (A1) who did not carry this variant used as another control alongside the normal control (lane C). We have also updated the figure legend to clearly label each lane with the sample ID, facilitating clearer interpretation (Kindly review the Track-changes version, page 13, lines 345-352).

  • Figure 4.The authors should detail what represent the violin plots in each graph and the statistical inferential test used, as well as the post-hoc tests used. The same should be described in the 4.6 Statistical Analysis paragraph.

  • Response: We thank the reviewer for this important observation. In response, we have updated the figure legend to clearly state that the violin plots represent the distribution and density of the respective cognitive and behavioral test scores (FSIQ, VIQ, NVIQ, Vineland, PPVT, and SRS) among affected and unaffected NR4A2 variant carriers. We also now specify that the statistical comparisons were conducted using one-way ANOVA followed by Tukey's multiple comparisons test for post-hoc analysis. These details have also been included in the revised Section 4.6 Statistical Analysis (Kindly review the Track-changes version, lines 385-392, and 659-662 (yellow highlighted)).

Materials and Methods

4.1 Study Cohort

When informing about the instruments used to appraise the intellectual functioning of patients, even if the authors reported the corresponding citation, it is always recommended to provide a brief description of each evaluation instrument employed, e.g. FSIQ, VIQ, NVIQ etc... This is important to make it immediately clear to readers, even those who are not experts in the field, about the type of evaluation carried out. This can increase the overall impact of the work.

  • Response: We agree with the reviewer and have revised Section 2.1 (General Demographic and Clinical Phenotypes) to include brief descriptions of the tools used, which helps ensuring clarity for readers unfamiliar with these psychometric tools (Kindly review the Track-changes version, lines 572-580 (yellow highlighted)).

4.2 Study Design

A more detailed description of the selection criteria should be provided. For instance, even if this is a genetic investigation, the authors should report if children presenting somatic disturbances other than those reported in Table 1, for instance atopic allergies or any other clinical condition that could impact the presenting phenotypes. The Approval of the Local Ethical Committee should be also added in this section, even if it is reported more precisely at the end of the manuscript.

  • Response: We appreciate this recommendation. The Study Design section has been expanded to provide more precise inclusion/exclusion criteria. Specifically, we now clarify that children with unrelated somatic conditions such as atopic allergies, chronic infections, or autoimmune diseases not affecting the CNS were excluded to avoid phenotypic confounders. In addition, although ethical approval was already mentioned at the end, it is now also explicitly stated within the Study Design section (Kindly review the Track-changes version, lines 586-594 (yellow highlighted)).

4.3 Genetic Analyses and Variant Validation

  • From “..paired-end reads. Reads were mapped….include putative variants in splicing regions”, the font is in a smaller size than the rest of the text.

  • Response: Thank you for your remark. The font size of the mentioned text was adjusted to fit the required format.

  • Figure 5. The authors should report in the picture, n= instead of n- ; the figure appears somehow unclear: it seems to contain already the results obtained as concerns the presence of the NR4A2  Please try to simplify this graph just reporting the schema of the different steps of the analysis.

  • Response: We acknowledge the reviewer’s suggestion and have modified Figure 5 accordingly: the annotation “n-” has been corrected to “n=”. Additionally, the visual has been simplified to display only the main analytical pipeline: patient recruitment, PBMC collection, exome sequencing, variant filtering, and functional annotation. The presence of NR4A2 variants has been removed from this flow to prevent redundancy with the results.
  • Please be advised that Figure 5 was relocated to section 2. Genetic Analysis: Frequency of De Novo NR4A2 Variants among ASD Children according to another reviewer’s suggestion, and therefore the figure number is updated to Figure 1, and the other figures’ numbers were updated accordingly.

  • The authors should provide a short description of the procedure of blood sampling and PBMC isolation. The same should be done for all other procedures.

  • Response: Thank you for this valuable point. We have added a concise description in the relevant section. Similar descriptions have been added for other experimental procedures where applicable (Kindly review the Track-changes version, page 19, lines 595-598 (yellow highlighted)).

4.6 Statistical Analysis

As indicated previously the authors should report the descriptive and inferential statistics used, with a short motivation.

  • Response: Thank you for your reminder. Section 4.6 Statistical Analysis was modified as per your previous suggestion (Kindly review the Track-changes version, lines 659-662 (yellow highlighted)).

Discussion

The discussion is well presented. Just maybe the authors should provide some hypotheses about the potential role of NR4A2 mutant protein in the ASD phenome. They should also discuss more the gene x environment effects in respect to the 3 variants present in non-affected and affected carriers. They should also encourage at the end of the discussion to prospect further deeper genomic investigations in a greater number of subjects through multi-center investigations; as well, proteomic and metabolomic analyses of these subjects should be developed, in order to elucidate the etiopathogenetic impact of these variants also at the molecular phenotype level.  

  • Response: We appreciate the reviewer’s supportive feedback and have now expanded the Discussion section to:

  • Highlight the potential role of NR4A2 mutant protein in the ASD phenome (Kindly review the Track-changes version, lines 403-433 (yellow highlighted)),

  • Discuss possible gene X environment interactions, including maternal metabolic status and environmental stressors that may influence NR4A2 expression (Kindly review the Track-changes version, lines 547-548, 554-556 (yellow highlighted)), and

  • Suggest that multi-omic approaches (genomic, proteomic, and metabolomic) in larger, multi-center cohorts will be essential for elucidating the mechanistic pathways of NR4A2-associated ASD (Kindly review the Track-changes version, lines 560-563 (yellow highlighted)).

  • The added text is supported with relevant references where applicable ( 23 – 30).

Reviewer 2 Report

Comments and Suggestions for Authors

The manuscript addresses the complex issue of NR4A2-related ASD  through  the clinical and molecular screening   of  338 ASD children from 315 unrelated families leading to the characterization of 10 de novo NR4A2 variants in 8 unrelated probands and 2 affected siblings from 8 unrelated families.  All  probands harbor multiple heterozygous NR4A2  variants, most of which are pathogenic loss-of-function variants, some recurrent among affected and healthy subjects.  This first investigated  Saudi Arabia cohort confirms that NR4A2 is a relevant gene in ASD  and  “clinical” expression of its pathogenic variants is influenced by several genetic, epigenetic and  environmental factors .

The clinical description of the cohort is complete and accurate (Table 1) and  Tables  2 and 3  are well structured and clear.  Conversely,  Figures 1 and 2 are  quite hard to read even upon magnification. The Authors should find a more schematic way  or select  images parts to present these figures  allowing the reader to capture the key  features of the selected  pathogenic variants .

Anticipating  Fig 5 as Fig.1 in the 2.2 section would help the reader to understand the relationship of patients and mutations ,

A weak point is the functional analysis of the splice region variant (CNV) harbored by 6 probands . The Authors do not comment the inherent Fig 3  which  legend does not  name the lanes  according to the  CNV carriers and their discordant and concordant siblings: the lack of the expected transcripts is clear, while the aberrant  transcripts are visible (original images ) in two samples of Panel B and only in two of the four samples of panel A . The results  appear  somehow preliminary .

The bibliography needs careful editing as the doi of several references are incorrect  (see  ref : 7,8, 9, 13, 22, 24 ,32, 34, 35, 38 etc) not allowing the access to the reader. Ref 4 (Russian) should be replaced.

The overall concerns  preclude to consider the manuscript in its present form                                                                                                                                                                                                                                                                                                                                                                                                                                                                                                                                                                                                                                                                                                                                                                                                                                                       

Minor : use Italics for the NR4A2 gene and its variants thorough the manuscript

Abstract line 36:  three variants affecting splicing

The manuscript addresses the complex issue of NR4A2-related ASD  through  the clinical and molecular screening   of  338 ASD children from 315 unrelated families leading to the characterization of 10 de novo NR4A2 variants in 8 unrelated probands and 2 affected siblings from 8 unrelated families.  All  probands harbor multiple NR4A2  variants, most of which are pathogenic loss-of-functio variants, some recurrent among affected and healthy subjects.  This first investigated  Saudi Arabia cohort confirms that NR4A2 is a relevant gene in ASD  and  “clinical” expression of its pathogenic  is influenced by  other genetic, epigenetic and  environmental factors .

The clinical description of the cohort is complete and accurate (Table 1) and  Tables  2 and 3  are well structured and clear.  Conversely,  Figures 1 and 2 are  quite hard to read even upon magnification. The authors should find a more schematic way to  allowing the reader to capture the key  features of the selected  pathogenic variants .

Anticipating  Fig 5 as Fig.1 in the 2.2 section would help the reader to understand the relationship of patients and mutations ,

A weak point is the functional analysis of the splice region variant (CNV) harbored by 6 probands . The Authors do not comment the inherent Fig 3  which  legend does not  name the lanes  according to the  CNV carriers and their discordant and concordant siblings: the lack of the expected transcripts is clear, while the aberrant  transcripts are visible (original images ) in two samples of Panel B and only in two of the four samples of panel A . The results  appear  somehow preliminary .

The bibliography needs careful editing as the doi of several references are incorrect  (see  ref : 7,8, 9, 13, 22, 24 ,32, 34, 35, 38 etc) not allowing the access to the reader. Ref 4 (Russian) should be replaced.

The overall concerns  preclude to consider the manuscript in its present form                                                                                                                                                                                                                                                                                                                                                                                                                                                                                                                                                                                                                                                                                                                                                                                                                                                         

Minor : use Italics for the NR4A2 gene and its variants thorough the manuscript

Abstract line 36:  three variants affecting splicing

The manuscript addresses the complex issue of NR4A2-related ASD  through  the clinical and molecular screening   of  338 ASD children from 315 unrelated families leading to the characterization of 10 de novo NR4A2 variants in 8 unrelated probands and 2 affected siblings from 8 unrelated families.  All  probands harbor multiple NR4A2  variants, most of which are pathogenic loss-of-functio variants, some recurrent among affected and healthy subjects.  This first investigated  Saudi Arabia cohort confirms that NR4A2 is a relevant gene in ASD  and  “clinical” expression of its pathogenic  is influenced by  other genetic, epigenetic and  environmental factors .

The clinical description of the cohort is complete and accurate (Table 1) and  Tables  2 and 3  are well structured and clear.  Conversely,  Figures 1 and 2 are  quite hard to read even upon magnification. The authors should find a more schematic way to  allowing the reader to capture the key  features of the selected  pathogenic variants .

Anticipating  Fig 5 as Fig.1 in the 2.2 section would help the reader to understand the relationship of patients and mutations ,

A weak point is the functional analysis of the splice region variant (CNV) harbored by 6 probands . The Authors do not comment the inherent Fig 3  which  legend does not  name the lanes  according to the  CNV carriers and their discordant and concordant siblings: the lack of the expected transcripts is clear, while the aberrant  transcripts are visible (original images ) in two samples of Panel B and only in two of the four samples of panel A . The results  appear  somehow preliminary .

The bibliography needs careful editing as the doi of several references are incorrect  (see  ref : 7,8, 9, 13, 22, 24 ,32, 34, 35, 38 etc) not allowing the access to the reader. Ref 4 (Russian) should be replaced.

The overall concerns  preclude to consider the manuscript in its present form                                                                                                                                                                                                                                                                                                                                                                                                                                                                                                                                                                                                                                                                                                                                                                                                                                                         

Minor : use Italics for the NR4A2 gene and its variants thorough the manuscript

Abstract line 36:  three variants affecting splicing

The manuscript addresses the complex issue of NR4A2-related ASD  through  the clinical and molecular screening   of  338 ASD children from 315 unrelated families leading to the characterization of 10 de novo NR4A2 variants in 8 unrelated probands and 2 affected siblings from 8 unrelated families.  All  probands harbor multiple NR4A2  variants, most of which are pathogenic loss-of-functio variants, some recurrent among affected and healthy subjects.  This first investigated  Saudi Arabia cohort confirms that NR4A2 is a relevant gene in ASD  and  “clinical” expression of its pathogenic  is influenced by  other genetic, epigenetic and  environmental factors .

The clinical description of the cohort is complete and accurate (Table 1) and  Tables  2 and 3  are well structured and clear.  Conversely,  Figures 1 and 2 are  quite hard to read even upon magnification. The authors should find a more schematic way to  allowing the reader to capture the key  features of the selected  pathogenic variants .

Anticipating  Fig 5 as Fig.1 in the 2.2 section would help the reader to understand the relationship of patients and mutations ,

A weak point is the functional analysis of the splice region variant (CNV) harbored by 6 probands . The Authors do not comment the inherent Fig 3  which  legend does not  name the lanes  according to the  CNV carriers and their discordant and concordant siblings: the lack of the expected transcripts is clear, while the aberrant  transcripts are visible (original images ) in two samples of Panel B and only in two of the four samples of panel A . The results  appear  somehow preliminary .

The bibliography needs careful editing as the doi of several references are incorrect  (see  ref : 7,8, 9, 13, 22, 24 ,32, 34, 35, 38 etc) not allowing the access to the reader. Ref 4 (Russian) should be replaced.

The overall concerns  preclude to consider the manuscript in its present form                                                                                                                                                                                                                                                                                                                                                                                                                                                                                                                                                                                                                                                                                                                                                                                                                                                         

Minor : use Italics for the NR4A2 gene and its variants thorough the manuscript

Abstract line 36:  three variants affecting splicing

The manuscript addresses the complex issue of NR4A2-related ASD  through  the clinical and molecular screening   of  338 ASD children from 315 unrelated families leading to the characterization of 10 de novo NR4A2 variants in 8 unrelated probands and 2 affected siblings from 8 unrelated families.  All  probands harbor multiple NR4A2  variants, most of which are pathogenic loss-of-functio variants, some recurrent among affected and healthy subjects.  This first investigated  Saudi Arabia cohort confirms that NR4A2 is a relevant gene in ASD  and  “clinical” expression of its pathogenic  is influenced by  other genetic, epigenetic and  environmental factors .

The clinical description of the cohort is complete and accurate (Table 1) and  Tables  2 and 3  are well structured and clear.  Conversely,  Figures 1 and 2 are  quite hard to read even upon magnification. The authors should find a more schematic way to  allowing the reader to capture the key  features of the selected  pathogenic variants .

Anticipating  Fig 5 as Fig.1 in the 2.2 section would help the reader to understand the relationship of patients and mutations ,

A weak point is the functional analysis of the splice region variant (CNV) harbored by 6 probands . The Authors do not comment the inherent Fig 3  which  legend does not  name the lanes  according to the  CNV carriers and their discordant and concordant siblings: the lack of the expected transcripts is clear, while the aberrant  transcripts are visible (original images ) in two samples of Panel B and only in two of the four samples of panel A . The results  appear  somehow preliminary .

The bibliography needs careful editing as the doi of several references are incorrect  (see  ref : 7,8, 9, 13, 22, 24 ,32, 34, 35, 38 etc) not allowing the access to the reader. Ref 4 (Russian) should be replaced.

The overall concerns  preclude to consider the manuscript in its present form                                                                                                                                                                                                                                                                                                                                                                                                                                                                                                                                                                                                                                                                                                                                                                                                                                                         

Minor : use Italics for the NR4A2 gene and its variants thorough the manuscript

Abstract line 36:  three variants affecting splicing

The manuscript addresses the complex issue of NR4A2-related ASD  through  the clinical and molecular screening   of  338 ASD children from 315 unrelated families leading to the characterization of 10 de novo NR4A2 variants in 8 unrelated probands and 2 affected siblings from 8 unrelated families.  All  probands harbor multiple NR4A2  variants, most of which are pathogenic loss-of-functio variants, some recurrent among affected and healthy subjects.  This first investigated  Saudi Arabia cohort confirms that NR4A2 is a relevant gene in ASD  and  “clinical” expression of its pathogenic  is influenced by  other genetic, epigenetic and  environmental factors .

The clinical description of the cohort is complete and accurate (Table 1) and  Tables  2 and 3  are well structured and clear.  Conversely,  Figures 1 and 2 are  quite hard to read even upon magnification. The authors should find a more schematic way to  allowing the reader to capture the key  features of the selected  pathogenic variants .

Anticipating  Fig 5 as Fig.1 in the 2.2 section would help the reader to understand the relationship of patients and mutations ,

A weak point is the functional analysis of the splice region variant (CNV) harbored by 6 probands . The Authors do not comment the inherent Fig 3  which  legend does not  name the lanes  according to the  CNV carriers and their discordant and concordant siblings: the lack of the expected transcripts is clear, while the aberrant  transcripts are visible (original images ) in two samples of Panel B and only in two of the four samples of panel A . The results  appear  somehow preliminary .

The bibliography needs careful editing as the doi of several references are incorrect  (see  ref : 7,8, 9, 13, 22, 24 ,32, 34, 35, 38 etc) not allowing the access to the reader. Ref 4 (Russian) should be replaced.

The overall concerns  preclude to consider the manuscript in its present form                                                                                                                                                                                                                                                                                                                                                                                                                                                                                                                                                                                                                                                                                                                                                                                                                                                         

Minor : use Italics for the NR4A2 gene and its variants thorough the manuscript

Abstract line 36:  three variants affecting splicing

The manuscript addresses the complex issue of NR4A2-related ASD  through  the clinical and molecular screening   of  338 ASD children from 315 unrelated families leading to the characterization of 10 de novo NR4A2 variants in 8 unrelated probands and 2 affected siblings from 8 unrelated families.  All  probands harbor multiple NR4A2  variants, most of which are pathogenic loss-of-functio variants, some recurrent among affected and healthy subjects.  This first investigated  Saudi Arabia cohort confirms that NR4A2 is a relevant gene in ASD  and  “clinical” expression of its pathogenic  is influenced by  other genetic, epigenetic and  environmental factors .

The clinical description of the cohort is complete and accurate (Table 1) and  Tables  2 and 3  are well structured and clear.  Conversely,  Figures 1 and 2 are  quite hard to read even upon magnification. The authors should find a more schematic way to  allowing the reader to capture the key  features of the selected  pathogenic variants .

Anticipating  Fig 5 as Fig.1 in the 2.2 section would help the reader to understand the relationship of patients and mutations ,

A weak point is the functional analysis of the splice region variant (CNV) harbored by 6 probands . The Authors do not comment the inherent Fig 3  which  legend does not  name the lanes  according to the  CNV carriers and their discordant and concordant siblings: the lack of the expected transcripts is clear, while the aberrant  transcripts are visible (original images ) in two samples of Panel B and only in two of the four samples of panel A . The results  appear  somehow preliminary .

The bibliography needs careful editing as the doi of several references are incorrect  (see  ref : 7,8, 9, 13, 22, 24 ,32, 34, 35, 38 etc) not allowing the access to the reader. Ref 4 (Russian) should be replaced.

The overall concerns  preclude to consider the manuscript in its present form                                                                                                                                                                                                                                                                                                                                                                                                                                                                                                                                                                                                                                                                                                                                                                                                                                                         

Minor : use Italics for the NR4A2 gene and its variants thorough the manuscript

Abstract line 36:  three variants affecting splicing

The manuscript addresses the complex issue of NR4A2-related ASD  through  the clinical and molecular screening   of  338 ASD children from 315 unrelated families leading to the characterization of 10 de novo NR4A2 variants in 8 unrelated probands and 2 affected siblings from 8 unrelated families.  All  probands harbor multiple NR4A2  variants, most of which are pathogenic loss-of-functio variants, some recurrent among affected and healthy subjects.  This first investigated  Saudi Arabia cohort confirms that NR4A2 is a relevant gene in ASD  and  “clinical” expression of its pathogenic  is influenced by  other genetic, epigenetic and  environmental factors .

The clinical description of the cohort is complete and accurate (Table 1) and  Tables  2 and 3  are well structured and clear.  Conversely,  Figures 1 and 2 are  quite hard to read even upon magnification. The authors should find a more schematic way to  allowing the reader to capture the key  features of the selected  pathogenic variants .

Anticipating  Fig 5 as Fig.1 in the 2.2 section would help the reader to understand the relationship of patients and mutations ,

A weak point is the functional analysis of the splice region variant (CNV) harbored by 6 probands . The Authors do not comment the inherent Fig 3  which  legend does not  name the lanes  according to the  CNV carriers and their discordant and concordant siblings: the lack of the expected transcripts is clear, while the aberrant  transcripts are visible (original images ) in two samples of Panel B and only in two of the four samples of panel A . The results  appear  somehow preliminary .

The bibliography needs careful editing as the doi of several references are incorrect  (see  ref : 7,8, 9, 13, 22, 24 ,32, 34, 35, 38 etc) not allowing the access to the reader. Ref 4 (Russian) should be replaced.

The overall concerns  preclude to consider the manuscript in its present form                                                                                                                                                                                                                                                                                                                                                                                                                                                                                                                                                                                                                                                                                                                                                                                                                                                         

Minor : use Italics for the NR4A2 gene and its variants thorough the manuscript

Abstract line 36:  three variants affecting splicing

The manuscript addresses the complex issue of NR4A2-related ASD  through  the clinical and molecular screening   of  338 ASD children from 315 unrelated families leading to the characterization of 10 de novo NR4A2 variants in 8 unrelated probands and 2 affected siblings from 8 unrelated families.  All  probands harbor multiple NR4A2  variants, most of which are pathogenic loss-of-functio variants, some recurrent among affected and healthy subjects.  This first investigated  Saudi Arabia cohort confirms that NR4A2 is a relevant gene in ASD  and  “clinical” expression of its pathogenic  is influenced by  other genetic, epigenetic and  environmental factors .

The clinical description of the cohort is complete and accurate (Table 1) and  Tables  2 and 3  are well structured and clear.  Conversely,  Figures 1 and 2 are  quite hard to read even upon magnification. The authors should find a more schematic way to  allowing the reader to capture the key  features of the selected  pathogenic variants .

Anticipating  Fig 5 as Fig.1 in the 2.2 section would help the reader to understand the relationship of patients and mutations ,

A weak point is the functional analysis of the splice region variant (CNV) harbored by 6 probands . The Authors do not comment the inherent Fig 3  which  legend does not  name the lanes  according to the  CNV carriers and their discordant and concordant siblings: the lack of the expected transcripts is clear, while the aberrant  transcripts are visible (original images ) in two samples of Panel B and only in two of the four samples of panel A . The results  appear  somehow preliminary .

The bibliography needs careful editing as the doi of several references are incorrect  (see  ref : 7,8, 9, 13, 22, 24 ,32, 34, 35, 38 etc) not allowing the access to the reader. Ref 4 (Russian) should be replaced.

The overall concerns  preclude to consider the manuscript in its present form                                                                                                                                                                                                                                                                                                                                                                                                                                                                                                                                                                                                                                                                                                                                                                                                                                                         

Minor : use Italics for the NR4A2 gene and its variants thorough the manuscript

Abstract line 36:  three variants affecting splicing

The manuscript addresses the complex issue of NR4A2-related ASD  through  the clinical and molecular screening   of  338 ASD children from 315 unrelated families leading to the characterization of 10 de novo NR4A2 variants in 8 unrelated probands and 2 affected siblings from 8 unrelated families.  All  probands harbor multiple NR4A2  variants, most of which are pathogenic loss-of-functio variants, some recurrent among affected and healthy subjects.  This first investigated  Saudi Arabia cohort confirms that NR4A2 is a relevant gene in ASD  and  “clinical” expression of its pathogenic  is influenced by  other genetic, epigenetic and  environmental factors .

The clinical description of the cohort is complete and accurate (Table 1) and  Tables  2 and 3  are well structured and clear.  Conversely,  Figures 1 and 2 are  quite hard to read even upon magnification. The authors should find a more schematic way to  allowing the reader to capture the key  features of the selected  pathogenic variants .

Anticipating  Fig 5 as Fig.1 in the 2.2 section would help the reader to understand the relationship of patients and mutations ,

A weak point is the functional analysis of the splice region variant (CNV) harbored by 6 probands . The Authors do not comment the inherent Fig 3  which  legend does not  name the lanes  according to the  CNV carriers and their discordant and concordant siblings: the lack of the expected transcripts is clear, while the aberrant  transcripts are visible (original images ) in two samples of Panel B and only in two of the four samples of panel A . The results  appear  somehow preliminary .

The bibliography needs careful editing as the doi of several references are incorrect  (see  ref : 7,8, 9, 13, 22, 24 ,32, 34, 35, 38 etc) not allowing the access to the reader. Ref 4 (Russian) should be replaced.

The overall concerns  preclude to consider the manuscript in its present form                                                                                                                                                                                                                                                                                                                                                                                                                                                                                                                                                                                                                                                                                                                                                                                                                                                         

Minor : use Italics for the NR4A2 gene and its variants thorough the manuscript

Abstract line 36:  three variants affecting splicing

The manuscript addresses the complex issue of NR4A2-related ASD  through  the clinical and molecular screening   of  338 ASD children from 315 unrelated families leading to the characterization of 10 de novo NR4A2 variants in 8 unrelated probands and 2 affected siblings from 8 unrelated families.  All  probands harbor multiple NR4A2  variants, most of which are pathogenic loss-of-functio variants, some recurrent among affected and healthy subjects.  This first investigated  Saudi Arabia cohort confirms that NR4A2 is a relevant gene in ASD  and  “clinical” expression of its pathogenic  is influenced by  other genetic, epigenetic and  environmental factors .

The clinical description of the cohort is complete and accurate (Table 1) and  Tables  2 and 3  are well structured and clear.  Conversely,  Figures 1 and 2 are  quite hard to read even upon magnification. The authors should find a more schematic way to  allowing the reader to capture the key  features of the selected  pathogenic variants .

Anticipating  Fig 5 as Fig.1 in the 2.2 section would help the reader to understand the relationship of patients and mutations ,

A weak point is the functional analysis of the splice region variant (CNV) harbored by 6 probands . The Authors do not comment the inherent Fig 3  which  legend does not  name the lanes  according to the  CNV carriers and their discordant and concordant siblings: the lack of the expected transcripts is clear, while the aberrant  transcripts are visible (original images ) in two samples of Panel B and only in two of the four samples of panel A . The results  appear  somehow preliminary .

The bibliography needs careful editing as the doi of several references are incorrect  (see  ref : 7,8, 9, 13, 22, 24 ,32, 34, 35, 38 etc) not allowing the access to the reader. Ref 4 (Russian) should be replaced.

The overall concerns  preclude to consider the manuscript in its present form                                                                                                                                                                                                                                                                                                                                                                                                                                                                                                                                                                                                                                                                                                                                                                                                                                                         

Minor : use Italics for the NR4A2 gene and its variants thorough the manuscript

Abstract line 36:  three variants affecting splicing

The manuscript addresses the complex issue of NR4A2-related ASD  through  the clinical and molecular screening   of  338 ASD children from 315 unrelated families leading to the characterization of 10 de novo NR4A2 variants in 8 unrelated probands and 2 affected siblings from 8 unrelated families.  All  probands harbor multiple NR4A2  variants, most of which are pathogenic loss-of-functio variants, some recurrent among affected and healthy subjects.  This first investigated  Saudi Arabia cohort confirms that NR4A2 is a relevant gene in ASD  and  “clinical” expression of its pathogenic  is influenced by  other genetic, epigenetic and  environmental factors .

The clinical description of the cohort is complete and accurate (Table 1) and  Tables  2 and 3  are well structured and clear.  Conversely,  Figures 1 and 2 are  quite hard to read even upon magnification. The authors should find a more schematic way to  allowing the reader to capture the key  features of the selected  pathogenic variants .

Anticipating  Fig 5 as Fig.1 in the 2.2 section would help the reader to understand the relationship of patients and mutations ,

A weak point is the functional analysis of the splice region variant (CNV) harbored by 6 probands . The Authors do not comment the inherent Fig 3  which  legend does not  name the lanes  according to the  CNV carriers and their discordant and concordant siblings: the lack of the expected transcripts is clear, while the aberrant  transcripts are visible (original images ) in two samples of Panel B and only in two of the four samples of panel A . The results  appear  somehow preliminary .

The bibliography needs careful editing as the doi of several references are incorrect  (see  ref : 7,8, 9, 13, 22, 24 ,32, 34, 35, 38 etc) not allowing the access to the reader. Ref 4 (Russian) should be replaced.

The overall concerns  preclude to consider the manuscript in its present form                                                                                                                                                                                                                                                                                                                                                                                                                                                                                                                                                                                                                                                                                                                                                                                                                                                         

Minor : use Italics for the NR4A2 gene and its variants thorough the manuscript

Abstract line 36:  three variants affecting splicing

The manuscript addresses the complex issue of NR4A2-related ASD  through  the clinical and molecular screening   of  338 ASD children from 315 unrelated families leading to the characterization of 10 de novo NR4A2 variants in 8 unrelated probands and 2 affected siblings from 8 unrelated families.  All  probands harbor multiple NR4A2  variants, most of which are pathogenic loss-of-functio variants, some recurrent among affected and healthy subjects.  This first investigated  Saudi Arabia cohort confirms that NR4A2 is a relevant gene in ASD  and  “clinical” expression of its pathogenic  is influenced by  other genetic, epigenetic and  environmental factors .

The clinical description of the cohort is complete and accurate (Table 1) and  Tables  2 and 3  are well structured and clear.  Conversely,  Figures 1 and 2 are  quite hard to read even upon magnification. The authors should find a more schematic way to  allowing the reader to capture the key  features of the selected  pathogenic variants .

Anticipating  Fig 5 as Fig.1 in the 2.2 section would help the reader to understand the relationship of patients and mutations ,

A weak point is the functional analysis of the splice region variant (CNV) harbored by 6 probands . The Authors do not comment the inherent Fig 3  which  legend does not  name the lanes  according to the  CNV carriers and their discordant and concordant siblings: the lack of the expected transcripts is clear, while the aberrant  transcripts are visible (original images ) in two samples of Panel B and only in two of the four samples of panel A . The results  appear  somehow preliminary .

The bibliography needs careful editing as the doi of several references are incorrect  (see  ref : 7,8, 9, 13, 22, 24 ,32, 34, 35, 38 etc) not allowing the access to the reader. Ref 4 (Russian) should be replaced.

The overall concerns  preclude to consider the manuscript in its present form                                                                                                                                                                                                                                                                                                                                                                                                                                                                                                                                                                                                                                                                                                                                                                                                                                                         

Minor : use Italics for the NR4A2 gene and its variants thorough the manuscript

Abstract line 36:  three variants affecting splicing

The manuscript addresses the complex issue of NR4A2-related ASD  through  the clinical and molecular screening   of  338 ASD children from 315 unrelated families leading to the characterization of 10 de novo NR4A2 variants in 8 unrelated probands and 2 affected siblings from 8 unrelated families.  All  probands harbor multiple NR4A2  variants, most of which are pathogenic loss-of-functio variants, some recurrent among affected and healthy subjects.  This first investigated  Saudi Arabia cohort confirms that NR4A2 is a relevant gene in ASD  and  “clinical” expression of its pathogenic  is influenced by  other genetic, epigenetic and  environmental factors .

The clinical description of the cohort is complete and accurate (Table 1) and  Tables  2 and 3  are well structured and clear.  Conversely,  Figures 1 and 2 are  quite hard to read even upon magnification. The authors should find a more schematic way to  allowing the reader to capture the key  features of the selected  pathogenic variants .

Anticipating  Fig 5 as Fig.1 in the 2.2 section would help the reader to understand the relationship of patients and mutations ,

A weak point is the functional analysis of the splice region variant (CNV) harbored by 6 probands . The Authors do not comment the inherent Fig 3  which  legend does not  name the lanes  according to the  CNV carriers and their discordant and concordant siblings: the lack of the expected transcripts is clear, while the aberrant  transcripts are visible (original images ) in two samples of Panel B and only in two of the four samples of panel A . The results  appear  somehow preliminary .

The bibliography needs careful editing as the doi of several references are incorrect  (see  ref : 7,8, 9, 13, 22, 24 ,32, 34, 35, 38 etc) not allowing the access to the reader. Ref 4 (Russian) should be replaced.

The overall concerns  preclude to consider the manuscript in its present form                                                                                                                                                                                                                                                                                                                                                                                                                                                                                                                                                                                                                                                                                                                                                                                                                                                         

Minor : use Italics for the NR4A2 gene and its variants thorough the manuscript

Abstract line 36:  three variants affecting splicing

The manuscript addresses the complex issue of NR4A2-related ASD  through  the clinical and molecular screening   of  338 ASD children from 315 unrelated families leading to the characterization of 10 de novo NR4A2 variants in 8 unrelated probands and 2 affected siblings from 8 unrelated families.  All  probands harbor multiple NR4A2  variants, most of which are pathogenic loss-of-functio variants, some recurrent among affected and healthy subjects.  This first investigated  Saudi Arabia cohort confirms that NR4A2 is a relevant gene in ASD  and  “clinical” expression of its pathogenic  is influenced by  other genetic, epigenetic and  environmental factors .

The clinical description of the cohort is complete and accurate (Table 1) and  Tables  2 and 3  are well structured and clear.  Conversely,  Figures 1 and 2 are  quite hard to read even upon magnification. The authors should find a more schematic way to  allowing the reader to capture the key  features of the selected  pathogenic variants .

Anticipating  Fig 5 as Fig.1 in the 2.2 section would help the reader to understand the relationship of patients and mutations ,

A weak point is the functional analysis of the splice region variant (CNV) harbored by 6 probands . The Authors do not comment the inherent Fig 3  which  legend does not  name the lanes  according to the  CNV carriers and their discordant and concordant siblings: the lack of the expected transcripts is clear, while the aberrant  transcripts are visible (original images ) in two samples of Panel B and only in two of the four samples of panel A . The results  appear  somehow preliminary .

The bibliography needs careful editing as the doi of several references are incorrect  (see  ref : 7,8, 9, 13, 22, 24 ,32, 34, 35, 38 etc) not allowing the access to the reader. Ref 4 (Russian) should be replaced.

The overall concerns  preclude to consider the manuscript in its present form                                                                                                                                                                                                                                                                                                                                                                                                                                                                                                                                                                                                                                                                                                                                                                                                                                                         

Minor : use Italics for the NR4A2 gene and its variants thorough the manuscript

Abstract line 36:  three variants affecting splicing

The manuscript addresses the complex issue of NR4A2-related ASD  through  the clinical and molecular screening   of  338 ASD children from 315 unrelated families leading to the characterization of 10 de novo NR4A2 variants in 8 unrelated probands and 2 affected siblings from 8 unrelated families.  All  probands harbor multiple NR4A2  variants, most of which are pathogenic loss-of-functio variants, some recurrent among affected and healthy subjects.  This first investigated  Saudi Arabia cohort confirms that NR4A2 is a relevant gene in ASD  and  “clinical” expression of its pathogenic  is influenced by  other genetic, epigenetic and  environmental factors .

The clinical description of the cohort is complete and accurate (Table 1) and  Tables  2 and 3  are well structured and clear.  Conversely,  Figures 1 and 2 are  quite hard to read even upon magnification. The authors should find a more schematic way to  allowing the reader to capture the key  features of the selected  pathogenic variants .

Anticipating  Fig 5 as Fig.1 in the 2.2 section would help the reader to understand the relationship of patients and mutations ,

A weak point is the functional analysis of the splice region variant (CNV) harbored by 6 probands . The Authors do not comment the inherent Fig 3  which  legend does not  name the lanes  according to the  CNV carriers and their discordant and concordant siblings: the lack of the expected transcripts is clear, while the aberrant  transcripts are visible (original images ) in two samples of Panel B and only in two of the four samples of panel A . The results  appear  somehow preliminary .

The bibliography needs careful editing as the doi of several references are incorrect  (see  ref : 7,8, 9, 13, 22, 24 ,32, 34, 35, 38 etc) not allowing the access to the reader. Ref 4 (Russian) should be replaced.

The overall concerns  preclude to consider the manuscript in its present form                                                                                                                                                                                                                                                                                                                                                                                                                                                                                                                                                                                                                                                                                                                                                                                                                                                         

Minor : use Italics for the NR4A2 gene and its variants thorough the manuscript

Abstract line 36:  three variants affecting splicing

The manuscript addresses the complex issue of NR4A2-related ASD  through  the clinical and molecular screening   of  338 ASD children from 315 unrelated families leading to the characterization of 10 de novo NR4A2 variants in 8 unrelated probands and 2 affected siblings from 8 unrelated families.  All  probands harbor multiple NR4A2  variants, most of which are pathogenic loss-of-functio variants, some recurrent among affected and healthy subjects.  This first investigated  Saudi Arabia cohort confirms that NR4A2 is a relevant gene in ASD  and  “clinical” expression of its pathogenic  is influenced by  other genetic, epigenetic and  environmental factors .

The clinical description of the cohort is complete and accurate (Table 1) and  Tables  2 and 3  are well structured and clear.  Conversely,  Figures 1 and 2 are  quite hard to read even upon magnification. The authors should find a more schematic way to  allowing the reader to capture the key  features of the selected  pathogenic variants .

Anticipating  Fig 5 as Fig.1 in the 2.2 section would help the reader to understand the relationship of patients and mutations ,

A weak point is the functional analysis of the splice region variant (CNV) harbored by 6 probands . The Authors do not comment the inherent Fig 3  which  legend does not  name the lanes  according to the  CNV carriers and their discordant and concordant siblings: the lack of the expected transcripts is clear, while the aberrant  transcripts are visible (original images ) in two samples of Panel B and only in two of the four samples of panel A . The results  appear  somehow preliminary .

The bibliography needs careful editing as the doi of several references are incorrect  (see  ref : 7,8, 9, 13, 22, 24 ,32, 34, 35, 38 etc) not allowing the access to the reader. Ref 4 (Russian) should be replaced.

The overall concerns  preclude to consider the manuscript in its present form                                                                                                                                                                                                                                                                                                                                                                                                                                                                                                                                                                                                                                                                                                                                                                                                                                                         

Minor : use Italics for the NR4A2 gene and its variants thorough the manuscript

Abstract line 36:  three variants affecting splicing

The manuscript addresses the complex issue of NR4A2-related ASD  through  the clinical and molecular screening   of  338 ASD children from 315 unrelated families leading to the characterization of 10 de novo NR4A2 variants in 8 unrelated probands and 2 affected siblings from 8 unrelated families.  All  probands harbor multiple NR4A2  variants, most of which are pathogenic loss-of-functio variants, some recurrent among affected and healthy subjects.  This first investigated  Saudi Arabia cohort confirms that NR4A2 is a relevant gene in ASD  and  “clinical” expression of its pathogenic  is influenced by  other genetic, epigenetic and  environmental factors .

The clinical description of the cohort is complete and accurate (Table 1) and  Tables  2 and 3  are well structured and clear.  Conversely,  Figures 1 and 2 are  quite hard to read even upon magnification. The authors should find a more schematic way to  allowing the reader to capture the key  features of the selected  pathogenic variants .

Anticipating  Fig 5 as Fig.1 in the 2.2 section would help the reader to understand the relationship of patients and mutations ,

A weak point is the functional analysis of the splice region variant (CNV) harbored by 6 probands . The Authors do not comment the inherent Fig 3  which  legend does not  name the lanes  according to the  CNV carriers and their discordant and concordant siblings: the lack of the expected transcripts is clear, while the aberrant  transcripts are visible (original images ) in two samples of Panel B and only in two of the four samples of panel A . The results  appear  somehow preliminary .

The bibliography needs careful editing as the doi of several references are incorrect  (see  ref : 7,8, 9, 13, 22, 24 ,32, 34, 35, 38 etc) not allowing the access to the reader. Ref 4 (Russian) should be replaced.

The overall concerns  preclude to consider the manuscript in its present form                                                                                                                                                                                                                                                                                                                                                                                                                                                                                                                                                                                                                                                                                                                                                                                                                                                         

Minor : use Italics for the NR4A2 gene and its variants thorough the manuscript

Abstract line 36:  three variants affecting splicing

The manuscript addresses the complex issue of NR4A2-related ASD  through  the clinical and molecular screening   of  338 ASD children from 315 unrelated families leading to the characterization of 10 de novo NR4A2 variants in 8 unrelated probands and 2 affected siblings from 8 unrelated families.  All  probands harbor multiple NR4A2  variants, most of which are pathogenic loss-of-functio variants, some recurrent among affected and healthy subjects.  This first investigated  Saudi Arabia cohort confirms that NR4A2 is a relevant gene in ASD  and  “clinical” expression of its pathogenic  is influenced by  other genetic, epigenetic and  environmental factors .

The clinical description of the cohort is complete and accurate (Table 1) and  Tables  2 and 3  are well structured and clear.  Conversely,  Figures 1 and 2 are  quite hard to read even upon magnification. The authors should find a more schematic way to  allowing the reader to capture the key  features of the selected  pathogenic variants .

Anticipating  Fig 5 as Fig.1 in the 2.2 section would help the reader to understand the relationship of patients and mutations ,

A weak point is the functional analysis of the splice region variant (CNV) harbored by 6 probands . The Authors do not comment the inherent Fig 3  which  legend does not  name the lanes  according to the  CNV carriers and their discordant and concordant siblings: the lack of the expected transcripts is clear, while the aberrant  transcripts are visible (original images ) in two samples of Panel B and only in two of the four samples of panel A . The results  appear  somehow preliminary .

The bibliography needs careful editing as the doi of several references are incorrect  (see  ref : 7,8, 9, 13, 22, 24 ,32, 34, 35, 38 etc) not allowing the access to the reader. Ref 4 (Russian) should be replaced.

The overall concerns  preclude to consider the manuscript in its present form                                                                                                                                                                                                                                                                                                                                                                                                                                                                                                                                                                                                                                                                                                                                                                                                                                                         

Minor : use Italics for the NR4A2 gene and its variants thorough the manuscript

Abstract line 36:  three variants affecting splicing

The manuscript addresses the complex issue of NR4A2-related ASD  through  the clinical and molecular screening   of  338 ASD children from 315 unrelated families leading to the characterization of 10 de novo NR4A2 variants in 8 unrelated probands and 2 affected siblings from 8 unrelated families.  All  probands harbor multiple NR4A2  variants, most of which are pathogenic loss-of-functio variants, some recurrent among affected and healthy subjects.  This first investigated  Saudi Arabia cohort confirms that NR4A2 is a relevant gene in ASD  and  “clinical” expression of its pathogenic  is influenced by  other genetic, epigenetic and  environmental factors .

The clinical description of the cohort is complete and accurate (Table 1) and  Tables  2 and 3  are well structured and clear.  Conversely,  Figures 1 and 2 are  quite hard to read even upon magnification. The authors should find a more schematic way to  allowing the reader to capture the key  features of the selected  pathogenic variants .

Anticipating  Fig 5 as Fig.1 in the 2.2 section would help the reader to understand the relationship of patients and mutations ,

A weak point is the functional analysis of the splice region variant (CNV) harbored by 6 probands . The Authors do not comment the inherent Fig 3  which  legend does not  name the lanes  according to the  CNV carriers and their discordant and concordant siblings: the lack of the expected transcripts is clear, while the aberrant  transcripts are visible (original images ) in two samples of Panel B and only in two of the four samples of panel A . The results  appear  somehow preliminary .

The bibliography needs careful editing as the doi of several references are incorrect  (see  ref : 7,8, 9, 13, 22, 24 ,32, 34, 35, 38 etc) not allowing the access to the reader. Ref 4 (Russian) should be replaced.

The overall concerns  preclude to consider the manuscript in its present form                                                                                                                                                                                                                                                                                                                                                                                                                                                                                                                                                                                                                                                                                                                                                                                                                                                         

Minor : use Italics for the NR4A2 gene and its variants thorough the manuscript

Abstract line 36:  three variants affecting splicing

The manuscript addresses the complex issue of NR4A2-related ASD  through  the clinical and molecular screening   of  338 ASD children from 315 unrelated families leading to the characterization of 10 de novo NR4A2 variants in 8 unrelated probands and 2 affected siblings from 8 unrelated families.  All  probands harbor multiple NR4A2  variants, most of which are pathogenic loss-of-functio variants, some recurrent among affected and healthy subjects.  This first investigated  Saudi Arabia cohort confirms that NR4A2 is a relevant gene in ASD  and  “clinical” expression of its pathogenic  is influenced by  other genetic, epigenetic and  environmental factors .

The clinical description of the cohort is complete and accurate (Table 1) and  Tables  2 and 3  are well structured and clear.  Conversely,  Figures 1 and 2 are  quite hard to read even upon magnification. The authors should find a more schematic way to  allowing the reader to capture the key  features of the selected  pathogenic variants .

Anticipating  Fig 5 as Fig.1 in the 2.2 section would help the reader to understand the relationship of patients and mutations ,

A weak point is the functional analysis of the splice region variant (CNV) harbored by 6 probands . The Authors do not comment the inherent Fig 3  which  legend does not  name the lanes  according to the  CNV carriers and their discordant and concordant siblings: the lack of the expected transcripts is clear, while the aberrant  transcripts are visible (original images ) in two samples of Panel B and only in two of the four samples of panel A . The results  appear  somehow preliminary .

The bibliography needs careful editing as the doi of several references are incorrect  (see  ref : 7,8, 9, 13, 22, 24 ,32, 34, 35, 38 etc) not allowing the access to the reader. Ref 4 (Russian) should be replaced.

The overall concerns  preclude to consider the manuscript in its present form                                                                                                                                                                                                                                                                                                                                                                                                                                                                                                                                                                                                                                                                                                                                                                                                                                                         

Minor : use Italics for the NR4A2 gene and its variants thorough the manuscript

Abstract line 36:  three variants affecting splicing

The manuscript addresses the complex issue of NR4A2-related ASD  through  the clinical and molecular screening   of  338 ASD children from 315 unrelated families leading to the characterization of 10 de novo NR4A2 variants in 8 unrelated probands and 2 affected siblings from 8 unrelated families.  All  probands harbor multiple NR4A2  variants, most of which are pathogenic loss-of-functio variants, some recurrent among affected and healthy subjects.  This first investigated  Saudi Arabia cohort confirms that NR4A2 is a relevant gene in ASD  and  “clinical” expression of its pathogenic  is influenced by  other genetic, epigenetic and  environmental factors .

The clinical description of the cohort is complete and accurate (Table 1) and  Tables  2 and 3  are well structured and clear.  Conversely,  Figures 1 and 2 are  quite hard to read even upon magnification. The authors should find a more schematic way to  allowing the reader to capture the key  features of the selected  pathogenic variants .

Anticipating  Fig 5 as Fig.1 in the 2.2 section would help the reader to understand the relationship of patients and mutations ,

A weak point is the functional analysis of the splice region variant (CNV) harbored by 6 probands . The Authors do not comment the inherent Fig 3  which  legend does not  name the lanes  according to the  CNV carriers and their discordant and concordant siblings: the lack of the expected transcripts is clear, while the aberrant  transcripts are visible (original images ) in two samples of Panel B and only in two of the four samples of panel A . The results  appear  somehow preliminary .

The bibliography needs careful editing as the doi of several references are incorrect  (see  ref : 7,8, 9, 13, 22, 24 ,32, 34, 35, 38 etc) not allowing the access to the reader. Ref 4 (Russian) should be replaced.

The overall concerns  preclude to consider the manuscript in its present form                                                                                                                                                                                                                                                                                                                                                                                                                                                                                                                                                                                                                                                                                                                                                                                                                                                         

Minor : use Italics for the NR4A2 gene and its variants thorough the manuscript

Abstract line 36:  three variants affecting splicing

The manuscript addresses the complex issue of NR4A2-related ASD  through  the clinical and molecular screening   of  338 ASD children from 315 unrelated families leading to the characterization of 10 de novo NR4A2 variants in 8 unrelated probands and 2 affected siblings from 8 unrelated families.  All  probands harbor multiple NR4A2  variants, most of which are pathogenic loss-of-functio variants, some recurrent among affected and healthy subjects.  This first investigated  Saudi Arabia cohort confirms that NR4A2 is a relevant gene in ASD  and  “clinical” expression of its pathogenic  is influenced by  other genetic, epigenetic and  environmental factors .

The clinical description of the cohort is complete and accurate (Table 1) and  Tables  2 and 3  are well structured and clear.  Conversely,  Figures 1 and 2 are  quite hard to read even upon magnification. The authors should find a more schematic way to  allowing the reader to capture the key  features of the selected  pathogenic variants .

Anticipating  Fig 5 as Fig.1 in the 2.2 section would help the reader to understand the relationship of patients and mutations ,

A weak point is the functional analysis of the splice region variant (CNV) harbored by 6 probands . The Authors do not comment the inherent Fig 3  which  legend does not  name the lanes  according to the  CNV carriers and their discordant and concordant siblings: the lack of the expected transcripts is clear, while the aberrant  transcripts are visible (original images ) in two samples of Panel B and only in two of the four samples of panel A . The results  appear  somehow preliminary .

The bibliography needs careful editing as the doi of several references are incorrect  (see  ref : 7,8, 9, 13, 22, 24 ,32, 34, 35, 38 etc) not allowing the access to the reader. Ref 4 (Russian) should be replaced.

The overall concerns  preclude to consider the manuscript in its present form                                                                                                                                                                                                                                                                                                                                                                                                                                                                                                                                                                                                                                                                                                                                                                                                                                                         

Minor : use Italics for the NR4A2 gene and its variants thorough the manuscript

Abstract line 36:  three variants affecting splicing

The manuscript addresses the complex issue of NR4A2-related ASD  through  the clinical and molecular screening   of  338 ASD children from 315 unrelated families leading to the characterization of 10 de novo NR4A2 variants in 8 unrelated probands and 2 affected siblings from 8 unrelated families.  All  probands harbor multiple NR4A2  variants, most of which are pathogenic loss-of-functio variants, some recurrent among affected and healthy subjects.  This first investigated  Saudi Arabia cohort confirms that NR4A2 is a relevant gene in ASD  and  “clinical” expression of its pathogenic  is influenced by  other genetic, epigenetic and  environmental factors .

The clinical description of the cohort is complete and accurate (Table 1) and  Tables  2 and 3  are well structured and clear.  Conversely,  Figures 1 and 2 are  quite hard to read even upon magnification. The authors should find a more schematic way to  allowing the reader to capture the key  features of the selected  pathogenic variants .

Anticipating  Fig 5 as Fig.1 in the 2.2 section would help the reader to understand the relationship of patients and mutations ,

A weak point is the functional analysis of the splice region variant (CNV) harbored by 6 probands . The Authors do not comment the inherent Fig 3  which  legend does not  name the lanes  according to the  CNV carriers and their discordant and concordant siblings: the lack of the expected transcripts is clear, while the aberrant  transcripts are visible (original images ) in two samples of Panel B and only in two of the four samples of panel A . The results  appear  somehow preliminary .

The bibliography needs careful editing as the doi of several references are incorrect  (see  ref : 7,8, 9, 13, 22, 24 ,32, 34, 35, 38 etc) not allowing the access to the reader. Ref 4 (Russian) should be replaced.

The overall concerns  preclude to consider the manuscript in its present form                                                                                                                                                                                                                                                                                                                                                                                                                                                                                                                                                                                                                                                                                                                                                                                                                                                         

Minor : use Italics for the NR4A2 gene and its variants thorough the manuscript

Abstract line 36:  three variants affecting splicing

The manuscript addresses the complex issue of NR4A2-related ASD  through  the clinical and molecular screening   of  338 ASD children from 315 unrelated families leading to the characterization of 10 de novo NR4A2 variants in 8 unrelated probands and 2 affected siblings from 8 unrelated families.  All  probands harbor multiple NR4A2  variants, most of which are pathogenic loss-of-functio variants, some recurrent among affected and healthy subjects.  This first investigated  Saudi Arabia cohort confirms that NR4A2 is a relevant gene in ASD  and  “clinical” expression of its pathogenic  is influenced by  other genetic, epigenetic and  environmental factors .

The clinical description of the cohort is complete and accurate (Table 1) and  Tables  2 and 3  are well structured and clear.  Conversely,  Figures 1 and 2 are  quite hard to read even upon magnification. The authors should find a more schematic way to  allowing the reader to capture the key  features of the selected  pathogenic variants .

Anticipating  Fig 5 as Fig.1 in the 2.2 section would help the reader to understand the relationship of patients and mutations ,

A weak point is the functional analysis of the splice region variant (CNV) harbored by 6 probands . The Authors do not comment the inherent Fig 3  which  legend does not  name the lanes  according to the  CNV carriers and their discordant and concordant siblings: the lack of the expected transcripts is clear, while the aberrant  transcripts are visible (original images ) in two samples of Panel B and only in two of the four samples of panel A . The results  appear  somehow preliminary .

The bibliography needs careful editing as the doi of several references are incorrect  (see  ref : 7,8, 9, 13, 22, 24 ,32, 34, 35, 38 etc) not allowing the access to the reader. Ref 4 (Russian) should be replaced.

The overall concerns  preclude to consider the manuscript in its present form                                                                                                                                                                                                                                                                                                                                                                                                                                                                                                                                                                                                                                                                                                                                                                                                                                                         

Minor : use Italics for the NR4A2 gene and its variants thorough the manuscript

Abstract line 36:  three variants affecting splicing

The manuscript addresses the complex issue of NR4A2-related ASD  through  the clinical and molecular screening   of  338 ASD children from 315 unrelated families leading to the characterization of 10 de novo NR4A2 variants in 8 unrelated probands and 2 affected siblings from 8 unrelated families.  All  probands harbor multiple NR4A2  variants, most of which are pathogenic loss-of-functio variants, some recurrent among affected and healthy subjects.  This first investigated  Saudi Arabia cohort confirms that NR4A2 is a relevant gene in ASD  and  “clinical” expression of its pathogenic  is influenced by  other genetic, epigenetic and  environmental factors .

The clinical description of the cohort is complete and accurate (Table 1) and  Tables  2 and 3  are well structured and clear.  Conversely,  Figures 1 and 2 are  quite hard to read even upon magnification. The authors should find a more schematic way to  allowing the reader to capture the key  features of the selected  pathogenic variants .

Anticipating  Fig 5 as Fig.1 in the 2.2 section would help the reader to understand the relationship of patients and mutations ,

A weak point is the functional analysis of the splice region variant (CNV) harbored by 6 probands . The Authors do not comment the inherent Fig 3  which  legend does not  name the lanes  according to the  CNV carriers and their discordant and concordant siblings: the lack of the expected transcripts is clear, while the aberrant  transcripts are visible (original images ) in two samples of Panel B and only in two of the four samples of panel A . The results  appear  somehow preliminary .

The bibliography needs careful editing as the doi of several references are incorrect  (see  ref : 7,8, 9, 13, 22, 24 ,32, 34, 35, 38 etc) not allowing the access to the reader. Ref 4 (Russian) should be replaced.

The overall concerns  preclude to consider the manuscript in its present form                                                                                                                                                                                                                                                                                                                                                                                                                                                                                                                                                                                                                                                                                                                                                                                                                                                         

Minor : use Italics for the NR4A2 gene and its variants thorough the manuscript

Abstract line 36:  three variants affecting splicing

The manuscript addresses the complex issue of NR4A2-related ASD  through  the clinical and molecular screening   of  338 ASD children from 315 unrelated families leading to the characterization of 10 de novo NR4A2 variants in 8 unrelated probands and 2 affected siblings from 8 unrelated families.  All  probands harbor multiple NR4A2  variants, most of which are pathogenic loss-of-functio variants, some recurrent among affected and healthy subjects.  This first investigated  Saudi Arabia cohort confirms that NR4A2 is a relevant gene in ASD  and  “clinical” expression of its pathogenic  is influenced by  other genetic, epigenetic and  environmental factors .

The clinical description of the cohort is complete and accurate (Table 1) and  Tables  2 and 3  are well structured and clear.  Conversely,  Figures 1 and 2 are  quite hard to read even upon magnification. The authors should find a more schematic way to  allowing the reader to capture the key  features of the selected  pathogenic variants .

Anticipating  Fig 5 as Fig.1 in the 2.2 section would help the reader to understand the relationship of patients and mutations ,

A weak point is the functional analysis of the splice region variant (CNV) harbored by 6 probands . The Authors do not comment the inherent Fig 3  which  legend does not  name the lanes  according to the  CNV carriers and their discordant and concordant siblings: the lack of the expected transcripts is clear, while the aberrant  transcripts are visible (original images ) in two samples of Panel B and only in two of the four samples of panel A . The results  appear  somehow preliminary .

The bibliography needs careful editing as the doi of several references are incorrect  (see  ref : 7,8, 9, 13, 22, 24 ,32, 34, 35, 38 etc) not allowing the access to the reader. Ref 4 (Russian) should be replaced.

The overall concerns  preclude to consider the manuscript in its present form                                                                                                                                                                                                                                                                                                                                                                                                                                                                                                                                                                                                                                                                                                                                                                                                                                                         

Minor : use Italics for the NR4A2 gene and its variants thorough the manuscript

Abstract line 36:  three variants affecting splicing

Author Response

First, we would like to express our gratitude to the respected editors and reviewers for their insightful comments on our manuscript. We were able to incorporate changes to reflect the suggestions provided, and we hope that we have given them the maximum consideration. Otherwise, we remain available for any other comments to improve the scientific quality of the manuscript. We have highlighted the changes within the Track Changes Version of the manuscript. Our point-by-point responses to the reviewer's comments and concerns are provided below.

Reviewer #2

The manuscript addresses the complex issue of NR4A2-related ASD through the clinical and molecular screening   of 338 ASD children from 315 unrelated families leading to the characterization of 10 de novo NR4A2 variants in 8 unrelated probands and 2 affected siblings from 8 unrelated families.  All probands harbor multiple heterozygous NR4A2 variants, most of which are pathogenic loss-of-function variants, some recurrent among affected and healthy subjects.  This first investigated Saudi Arabia cohort confirms that NR4A2 is a relevant gene in ASD and “clinical” expression of its pathogenic variants is influenced by several genetic, epigenetic and environmental factors.

  • The clinical description of the cohort is complete and accurate (Table 1) and Tables 2 and 3 are well structured and clear.  Conversely, Figures 1 and 2 are quite hard to read even upon magnification. The Authors should find a more schematic way or select images parts to present these figures allowing the reader to capture the key features of the selected pathogenic variants.
  • Response: We thank the respected reviewer for the insightful comment. Please be advised that the original figures have high resolution that becomes significantly reduced upon insertion in docx files. We attached the high-resolution figures in the revised files, and 2 is subdivided into Fig. 2A, Fig. 2B, and Fig. 2C to enhance the clarity of inserted figure as much as possible.
  • Anticipating Fig 5 as Fig.1 in the 2.2 section would help the reader to understand the relationship of patients and mutations,
  • Response: Thank you for your valuable suggestion. Done. The figure numbers are updated accordingly.
  • A weak point is the functional analysis of the splice region variant (CNV) harbored by 6 probands. The Authors do not comment the inherent Fig 3  which  legend does not  name the lanes  according to the  CNV carriers and their discordant and concordant siblings: the lack of the expected transcripts is clear, while the aberrant  transcripts are visible (original images ) in two samples of Panel B and only in two of the four samples of panel A . The results appear somehow preliminary.
  • Response: Thank you for your insight. Fig 4 (previously Fig. 3) is now updated to include the names of probands carrying the CNV and their concordant/discordant siblings (if any), as well as one proband (A1) who did not carry this variant used as another control alongside the normal control (lane C). We have also updated the figure legend to clearly label each lane with the sample ID, facilitating clearer interpretation (Kindly review the Track-changes version, lines 345-351 (green highlighted)).
  • We acknowledge that these results are preliminary and derived from a limited sample set and semi-quantitative RT-PCR analysis. Nonetheless, the absence of the expected transcript in all CNV carriers supports the pathogenic potential of this variant. Future work employing quantitative RNA sequence analysis or long-read transcriptome sequencing is warranted to further characterize the nature and consequence of the splicing disruption. These revisions have been incorporated in the manuscript (Kindly review the Track-changes version, lines 333-337; lines 352-354, and lines 691-693 (green highlighted)).

  • The bibliography needs careful editing as the doi of several references are incorrect (see  ref : 7,8, 9, 13, 22, 24 ,32, 34, 35, 38 etc) not allowing the access to the reader. Ref 4 (Russian) should be replaced.
  • Response: Thank you for your reminder. The doi links of all references were re-checked, and those mentioned along with others were Corrected. Ref 4 is replaced (Kindly review the Track-changes version, updated references are green highlighted)
  • Minor: use Italics for the NR4A2gene and its variants thorough the manuscript

  • Response: Thank you for your reminder. Done. (Kindly review the Track-changes version (cyan highlighted)).

  • Abstract line 36:  three variants affecting splicing
  • Response: Thank you for your suggestion. Corrected. (Kindly review the Track-changes version, green highlighted).

Round 2

Reviewer 2 Report

Comments and Suggestions for Authors

The manuscript addresses the complex issue of NR4A2-related ASD  through  the clinical and molecular screening   of  338 ASD children from 315 unrelated families leading to the characterization of 10 de novo NR4A2 variants in 8 unrelated probands and 2 affected siblings from 8 unrelated families.  All  probands harbor multiple NR4A2  variants, most of which are pathogenic loss-of-functio variants, some recurrent among affected and healthy subjects.  This first investigated  Saudi Arabia cohort confirms that NR4A2 is a relevant gene in ASD  and  “clinical” expression of its pathogenic  is influenced by  other genetic, epigenetic and  environmental factors .

The clinical description of the cohort is complete and accurate (Table 1) and  Tables  2 and 3  are well structured and clear.  Conversely,  Figures 1 and 2 are  quite hard to read even upon magnification. The authors should find a more schematic way to  allowing the reader to capture the key  features of the selected  pathogenic variants .

Anticipating  Fig 5 as Fig.1 in the 2.2 section would help the reader to understand the relationship of patients and mutations ,

A weak point is the functional analysis of the splice region variant (CNV) harbored by 6 probands . The Authors do not comment the inherent Fig 3  which  legend does not  name the lanes  according to the  CNV carriers and their discordant and concordant siblings: the lack of the expected transcripts is clear, while the aberrant  transcripts are visible (original images ) in two samples of Panel B and only in two of the four samples of panel A . The results  appear  somehow preliminary .

The bibliography needs careful editing as the doi of several references are incorrect  (see  ref : 7,8, 9, 13, 22, 24 ,32, 34, 35, 38 etc) not allowing the access to the reader. Ref 4 (Russian) should be replaced.

The overall concerns  preclude to consider the manuscript in its present form                                                                                                                                                                                                                                                                                                                                                                                                                                                                                                                                                                                                                                                                                                                                                                                                                                                         

Minor : use Italics for the NR4A2 gene and its variants thorough the manuscript

Abstract line 36:  three variants affecting splicing

Author Response

(The authors gave the same response as above.)

Round 3

Reviewer 2 Report

Comments and Suggestions for Authors

The incorporation in the revised version of a few raised suggestions is appreciated, but the following concerns remain:

Major:  Figures 2:  electropherograms  of  NR4A2  DNA from  different ASD probands showing heterozygous recurrent variants are hardly intellegible by the reader  including expert geneticist:  convert the information provided by sequencing  in a schema of the gene ( N-ter) with exons and introns drawn appropriately  and locate stars for exonic, intronic, splice site and CNV  pathogenic variants each correctly defined . The legend should be compacted  (Ex: an intronic  indel ( c-2-2del) located in intron 2 should be an intron 2 indel (c-2-2del))  and only inform on what A), B), C) , D represent.  A; proband A1 (according to Table 2)…… B) five probands harboring the nonsense variant c.44_45insA (p.S16*) concurrent with frameshift/splice-acceptor loss CNV………C) the mechanism of CNV generation is  complex to explain , especially as part of a multipanel figure……D) Recurrence of c.44_45insA (p.S16*) with another nonsense variant and E) recurrence of the CNV with a missense and a nonsense variant are clearly shown in Table 2.  

The same notes apply to Fig 3 , which should be schematized too: DNA sequencing of each ASD proband, their parents and their concordant/discordant siblings is a vauable opportunity yielding results; however the Figures are decorative but  useless  as nobody can catch the relevant sequence changes.

Minor: use  Italics when you mention the gene, not the protein (as at lines 79, 83, 85 87)

line 188    three at splice regions

line 198  that included an indel (c.-2-2 del) located in intron 2

line 200 both expected (delete of which)

lines  209 and 249  p,Q13= : delete=

line 213 delete were ; both expected

lines 311-313  rephrase this tortuous redundant sentence

 line 318 replace validated with supported

line 339 delete normal before splicing

line 488 replace associations with correlations

 References 11 and 13 have incorrect DOI

Author Response

We thank the reviewer for these insightful suggestions. In accordance with the reviewer’s valuable feedback, we were able to incorporate changes to reflect the suggestions provided. We have highlighted the changes within the Track Changes Version of the manuscript. Our point-by-point responses to the reviewer's comments and concerns are provided below.

Reviewer #2

  • Major:Figures 2:  Electropherogram  of  NR4A2 DNA from  different ASD probands showing heterozygous recurrent variants are hardly intelligible by the reader  including expert geneticist:  convert the information provided by sequencing  in a schema of the gene ( N-ter) with exons and introns drawn appropriately  and locate stars for exonic, intronic, splice site and CNV  pathogenic variants each correctly defined . The legend should be compacted  (Ex: an intronic  indel ( c-2-2del) located in intron 2 should be an intron 2 indel (c-2-2del))  and only inform on what A), B), C) , D represent.  A; proband A1 (according to Table 2)…… B) five probands harboring the nonsense variant c.44_45insA (p.S16*) concurrent with frameshift/splice-acceptor loss CNV………C) the mechanism of CNV generation is complex to explain, especially as part of a multipanel figure……D) Recurrence of c.44_45insA (p.S16*) with another nonsense variant and E) recurrence of the CNV with a missense and a nonsense variant are clearly shown in Table 2.  The same notes apply to Fig 3, which should be schematized too: DNA sequencing of each ASD proband, their parents and their concordant/discordant siblings is a vauable opportunity yielding results; however the Figures are decorative but useless as nobody can catch the relevant sequence changes.
  • Response: We thank the respected reviewer for the insightful suggestion. However, and considering that the variant type/distribution are fully described in the text and Table 2, the detailed variant sequences have been moved to a supplementary figures (S1 – S4), as they are less useful in the main figure, and therefore Figure 2 has been removed, and the figure numbers were updated accordingly.
  • We also appreciate the reviewer’s candor regarding Figure 3. In response, we have completely redesigned Figure 2 (previously Figure 3) to present a trio-based gene model for each quad family, and the recurrent/de novo variants identified in the probands belonging to simplex families (A3 and A11) along with their discordant siblings.
  • Minor: use Italics when you mention the gene, notthe protein (as at lines 79, 83, 85 87).
  • Response: Thank you for your reminder. Corrected.
  • line 188:    three atsplice regions
  • Response: Thank you for your reminder. Corrected.
  • line 198:  that included an indel (c.-2-2 del) located in intron 2
  • Response: Thank you for your suggestion. Corrected.
  • line 200: both expected (delete of which)

  • Response: Thank you for your reminder. Done.

  • lines  209 and 249:  p,Q13= : delete=
  • Response: Thank you for your suggestion. Done.
  • line 213: delete were ; both expected
  • Response: Thank you for your suggestion. Done.
  • line 213: delete were ; both expected
  • Response: Thank you for your suggestion. Done.
  • lines 311-313:  rephrase this tortuous redundant sentence
  • Response: Thank you for your insight. Lines 311-320 were rephrased to clearly describe how RT-PCR and targeted PCR amplification confirmed the absence of the canonical transcript in CNV-positive individuals (green highlighted).
  • line 318 replace validated with supported
  • Response: Thank you for your suggestion. Text from line 311 to line 320 was rephrased as per your previous suggestion.
  • line 339: delete normal before splicing
  • Response: Thank you for your suggestion. I think the respected reviewer refers to line 319. The word “normal” was removed.
  • line 488 replace associations with correlations
  • Response: Thank you for your suggestion. Done.
  • line 488 replace associations with correlations
  • Response: Thank you for your suggestion. Done.
  • References 11 and 13 have incorrect DOI
  • Response: Thank you for your reminder. Corrected.
